# State of the Art and New Trends from the 2022 Gism Annual Meeting

**DOI:** 10.3390/ijms24108902

**Published:** 2023-05-17

**Authors:** Ivana Ferrero, Camilla Francesca Proto, Alessia Giovanna Santa Banche Niclot, Elena Marini, Luisa Pascucci, Filippo Piccinini, Katia Mareschi

**Affiliations:** 1Stem Cell Transplantation and Cellular Therapy Laboratory, Paediatric Onco-Haematology Division, Regina Margherita Childrens’ Hospital, City of Health and Science of Turin, 10126 Turin, Italy; ivana.ferrero@unito.it; 2Department of Public Health and Paediatrics, University of Turin, 10126 Turin, Italyalessiagiovannasanta.bancheniclot@unito.it (A.G.S.B.N.); elena.marini@edu.unito.it (E.M.); 3Department of Veterinary Medicine, University of Perugia, 06125 Perugia, Italy; 4IRCCS Istituto Romagnolo per lo Studio dei Tumori (IRST) “Dino Amadori”, 47014 Meldola, Italy; 5Department of Medical and Surgical Sciences (DIMEC), University of Bologna, 40136 Bologna, Italy

**Keywords:** mesenchymal stem cells, annual meeting, Italian Mesenchymal Stem Cell Group (GISM), conference report

## Abstract

The 2022 Italian Mesenchymal Stem Cell Group (Gruppo Italiano Staminali Mesenchimali, GISM) Annual Meeting took place on 20–21 October 2022 in Turin (Italy), with the support of the University of Turin and the City of Health and Science of Turin. The novelty of this year’s meeting was its articulation, reflecting the new structure of GISM based on six sections: (*1*) Bringing advanced therapies to the clinic: trends and strategies, (*2*) GISM Next Generation, (*3*) New technologies for 3D culture systems, (*4*) Therapeutic applications of MSC-EVs in veterinary and human medicine, (*5*) Advancing MSC therapies in veterinary medicine: present challenges and future perspectives, (*6*) MSCs: a double-edged sword: friend or foe in oncology. National and international speakers presented their scientific works with the aim of promoting an interactive discussion and training for all attendees. The atmosphere was interactive, where ideas and questions between younger researchers and senior mentors were shared in all moments of the congress.

## 1. Introduction

The 2022 Italian Mesenchymal Stem Cell Group (Gruppo Italiano Staminali Mesenchimali, GISM) annual meeting took place on 20–21 October 2022 in Turin, with the support of the University of Turin and the City of Health and Science of Turin. The novelty of this year’s meeting was its articulation reflecting the new structure of GISM, based on six sections, that have been implemented during the last few years to better cover the interests concerning the different aspects of mesenchymal stem/stromal cell (MSC) use. The meeting brought together *GISM-Regenerative Medicine* with the support of the Cell Factories and researchers close to the clinic, and the *GISM-Next-Generation* section reflects the willingness of young talented researchers on MSCs. The *GISM-Veterinary* section, a very active group that reached not only researchers but also clinical veterinarians, introduced research in animal models towards the *GISM-Secretome*, focused on paracrine mechanisms mediated by soluble molecules and extracellular vesicles of MSCs. The last section, *GISM-Oncology*, covered the discussed relations between MSCs and tumours as well as the use of MSCs as a tool for drug delivery in cancer.

National and international speakers presented their scientific works with the aim of promoting an interactive discussion and training the youngest researchers.

## 2. Session Summary

The meeting was held in six different sessions in order to diversify the various study and research sectors (clinical, cell cultures and extra vesicles, veterinary, oncology, etc.) and to broaden the discussions and experiences of clinicians and researchers in the specific fields of study.

### 2.1. Session 1: “Bringing Advanced Therapies to the Clinic: Trends and Strategies”

The first session started with Graziella Pellegrini (University of Modena and Reggio Emilia, Italy) who presented her work about ex vivo expanded autologous human corneal epithelial cells containing stem cells and their production process for marketing authorisation (Holoclar) [1]. She and her group have been studying Limbal Stem Cell Deficiency (LSCD) since 1997 [2]. Since the regulatory guidelines changed in 2008 [3], their focus was aimed to obtain EMA approval for Holoclar which happened in 2015 [4] with a confirmatory trial of 80 patients. Their follow-up from 1997 to 2008 made it possible to have this court and, therefore, to already have in vivo studies, which allowed the facilitation of the approval. This allowed for real in vivo models as no animal eye can be compared to the human eye. She explained their procedure consisting of the isolation of epithelial cells from a very small biopsy from the healthy eye and their expansion and freezing. After thawing, the cells were transplanted with a special carrier on the damaged cornea [5]. The patients had a fast recovery with re-epithelialization after one week and re-vascularization after a couple of months with restored vision and stability. The final costs showed that Holoclar is less expensive than surgery (O-01).

Following, Stefano Cosma (University of Modena and Reggio Emilia, Italy) spoke about economic aspects and how researchers can bridge the funding gap. Scientific and operational factors, as well as economic factors (such as a lack or inadequacy of funding) represent risks of failure in medical research. Cosma and colleagues collected from the ClinicaTrial.gov database: 12,934 interventional studies were performed in Italy and 1137 in Europe, which were classified by keywords within advanced therapy medicinal product (ATMP) research. Through manual data reprocessing, they highlighted two important features: first, the information about failed and successful projects and their failure causes; second, the classification of the studies according to the funding source. An intersection of these data explained how financial resource allocations are correlated to the success or failure of clinical trials. This intervention was very important and elucidating, showing the point of view of an economist to understand and have indications of the economic aspects of the ATMPs (O-02).

After a break, Nicholas Crippa Orlandi (University of Siena, Italy) presented three cases of different body segments obtained through orthopaedic surgery that benefit from the use of gelled preparations containing autologous bone marrow MSCs. Crippa Orlandi and colleagues optimised the isolation and expansion protocol of bone marrow MSCs using Platelet Rich Plasma (PRP) instead of Foetal Bovine Serum (FBS), characterising cells seeded on the bovine bone matrix (SmartBone) or on a lyophilized acellular matrix, and they increased cell differentiation and osteogenic properties by adding growth factors. On SmartBone, the cells differentiated into osteoblasts and produced collagen. These data suggest that MSCs can provide valid help in complicated orthopaedic surgeries (O-03).

In order to have an overall view of the various sources of MSCs, Ilaria Roato (University of Turin, Italy) compared MSCs isolated from different districts of the oral cavity [6]. Adult Dental Pulp Stem Cells (DPSCs) are less osteogenic than Buccal Fat Pad Stem Cells (BFPSCs) but are more indicated for the pulp–dentin complex regeneration. MSCs isolated from human exfoliated teeth (SHED), DPSCs, and periodontal ligament stem cells (PDLSCs) showed no differences in the MSC marker immunophenotype, but SHED and DPSCs had a higher percentage of endothelial precursors than PDLSCs. The capability of MSCs to differentiate into a specific tissue depends also on the different anatomical origins, therefore, MSCs isolated from the oral cavity might be more effective than MSCs isolated from other origins such as adipose-derived stem cells (ASCs). Furthermore, these cells can be harvested after tooth extraction without causing pain or additional manoeuvres to the patient (O-04).

To conclude the first session of the congress, Marta Nardini (University of Genoa, Italy), selected as an oral communication, underlined the importance of cell behaviour after transplant because the distribution, engraftment, viability, and activity of transplanted cells are unclear. Nardini and colleagues developed a potency assay using Optoacoustic Imaging (OAI) in conjunction with rapid Gold-Nanostars (GNs). They characterised in vitro the complexes of GNs-labelled MSCs and extracellular vesicles (EVs), studied their distribution in sundry organs after a systemic injection using Multispectral Optoacoustic Tomography (MSOT), and tested in vivo toxicity. They isolated MSCs from bone marrow, cultured them with the addition of Platelet Lysate (PL) instead of FBS, and isolated the EVs from these cells. They injected labelled and unlabelled complexes into NuNu mice, and they tested in vivo toxicity at different time points with blood tests and histological analysis of the organs. GNs cannot overcome the Blood–Brain Barrier; after one day, they reached all organs and remained for all monitoring periods without any alteration, leading to a new perspective for tracking cell distribution (O-05).

### 2.2. Session 2: “Gism—Next Generation”

This section was organised by young GISM members and was designed to improve general skills around publishing, writing patents, and fundraising. The session started with Filippo Piccinini (IRCCS IRST ‘‘Dino Amadori”, Meldola, Italy, and University of Bologna, Italy). His presentation had a very catchy title, “Scientific articles: how to choose the right journal taking advantage of some opportunities”. The main goal was to discuss how to speed up the publication process, defining the best journal for a specific article and increasing the chances of publication by exploiting the opportunities that technology and social networks today provide. In particular, he introduced the most popular freely available journal finders and with a case study, he showed the audience how to exploit them. Finally, he practically described the meaning of the journal IF (impact factor) and the journal quartiles for a better understanding of the non-written rules behind the publication process for increasing the chances of a successful submission (O-06).

Next, Paola Bagnoli (IRCCS Ospedale Galeazzi Sant’Ambrogio, Milan, Italy) proceeded the section with a presentation related to the other side of research: “Technology transfer and intellectual property”. Besides summarising the main steps for patenting, she described the fundamental role of the Technology Transfer Offices (TTO) in scouting results, assisting researchers in protecting intellectual property, and activating stakeholders’ virtuous paths for the development of a project’s TRL (Technology Readiness Level) (O-07).

Finally, the “GISM—Next Generation” section ended with a round table composed of senior researchers working at different institutions, in particular, Maddalena Mastrogiacomo (University of Genoa, Italy), Luca Battistelli (IRCCS IRST Meldola, Italy), Enrico Lucarelli (IRCCS IOR Bologna, Italy), and Roberta Visone (Politecnico of Milan, Italy). The discussion was about difficulties and opportunities for fundraising ideas for MSC-related research. Despite the general difficulties claimed by all the speakers, they gave the audience several hints for future submissions. In particular, they cited several MSC-specific calls for applications mainly dedicated to young researchers.

### 2.3. Session 3: “New Technologies for 3D Culture Systems”

During this panel of the GISM congress, four innovative presentations illustrated new culture methods that could improve the expansion of mesenchymal cells for different purposes.

The first presentation was by Matteo Moretti (I.R.C.C.S Istituto Ortopedico Galeazzi of Milan, Italy), a bioengineer who presented a joint-on-a-chip 3D model [7]. This model was made by carefully layering chondrocytes, synovial fibroblasts, and synovial fluid in order to recreate on a small scale what is normally present in a patient and was used to test the effects of MSCs on osteoarthritis (OA) which is an inflammatory disease of the musculoskeletal system. In order to recreate the major hallmarks of this disease, injections of OA synovial fluid on the joint-on-chip were conducted and then the effect of MSCs was tested as a possible therapy option. These types of models could allow the personalization of the therapy for patients as the cells can be obtained directly from them and could allow the testing of innovative therapies (O-08).

Afterwards, the word was given to Maria Harmati (Biological Research Centre, Eötvös Lorand Research Network, Szeged, Hungary) who illustrated the crosstalk routes between tumour cells and stromal cells and vice versa. The crosstalk was conducted through extracellular vesicles tracked with specific dyes. She conducted her experiments not only on human ductal carcinoma cells but also on melanoma and osteosarcoma models. This provided more proof of the crosstalk that occurs and it provided proof on how the crosstalk changes in different tumour models (O-09).

Subsequently, Lucia Ceresa (Charles River Microbial Solutions, Italy) illustrated the suitability of Rapid Microbial Methods as a method to test the quality and safety of new cell-based medicinal products, using an ATP-based luminescent platform such as the Celsis platform which depletes the presence of cellular ATP and also allows fast detection of microbial presence (O-10).

Concluding this session, Silvia Scaglione (React4life startup, University of Genova, Italy) was chosen to discuss her poster as an oral communication. She presented her results on a novel Multi In Vivo Organ (MIVO) on a chip platform and how it could facilitate drug testing on cancer models [8]. Indeed, the models proposed were a 3D ovarian model developed and treated with Cisplatin, and a 3D neuroblastoma cancer model which was used to coculture with immune cells. These studies obtained a relevant disease model that can be used to investigate crosstalk between healthy and pathogenic cells and can also be employed as a drug screening platform (O-11).

### 2.4. Session 4: “Therapeutic Applications of MSC-EVs in Veterinary and Human Medicine”

Session 4 of the congress had a focus on the therapeutic applications of MSC-EVs in veterinary and human medicine. Stefania Bruno (University of Turin, Italy) presented the obstacles that can be found during the transfer of therapy with MSC-EVs from the laboratory to the clinic. In fact, MSC-EVs have been shown to have therapeutic effects in preclinical models of several diseases as they have pro-regenerative capacities and are considered therapeutic tools for various pathologies. To this end, EV characterisation is an area of intense investigation. To successfully translate EV research from the laboratory to patients, strategies must be designed to clinically test the safety and efficacy of MSC-EVs. Currently, manufacturing of cells and EVs, and quality controls, are being used for clinical testing of the feasibility of non-industrial processes. However, defining the mode of action in different diseases is essential for the MSC-EVs translation from the laboratory to the clinical setting (O-12).

In this regard, Silvia Zia (StemSel srl, Bologna, Italy) presented the Celector^®^ instrument, useful for ATMPs quality control and standardisation. Since MSCs are a heterogeneous population, the identity/purity of the cell population and its yield are critical issues that may limit their clinical application. To improve the characterisation of MSCs and standardise protocols, it is necessary to develop functional assays that evaluate the biophysical properties, profile, and quality of the cells. The Celector^®^ instrument is able to analyse, discriminate, and tag free separate a wide range of cells based on their physical characteristics, with high resolution and without damage. She explained the separation method, imaging acquisition, post-processing, and data analysis underlining that Celector@ is able to highlight physical differences related to cell viability and regenerative potential [9] (O-13).

Afterwards, space was given to young researchers to orally communicate their posters.

The first was the researcher Elena Ceccotti (University of Turin, Italy), who presented her work on chronic kidney diseases. In particular, she explained that human liver stem cells extracellular vesicles (HLSC-EVs) can be used as carriers for the transfer of active biological drugs both in vitro and in vivo models of renal ischemic reperfusion injury (IRI) associated with acute kidney injury (AKI) and chronic kidney disease (CKD). In this study, the results showed that in AKI mice, EV treatment attenuated kidney damage by reducing tubular necrosis and increasing tubular cell proliferation, and downregulated expression levels of fibrosis-related genes. In CKD mice, interstitial fibrosis and the expression levels of pro-fibrotic and pro-inflammatory genes were decreased [10]. Thus, the administration of HLSC-EVs immediately after renal IRI protects the kidney from the development of AKI and interferes with the development of subsequent CKD (O-14).

Proceeding with the session, Tarlan Eslami Arshaghi (University of Galway, Ireland) introduced the Aptamer approach: a fluorescence polarisation-based approach for EV quantification. This approach aimed to develop a novel high-throughput EV quantification tool based on the interaction between a fluorescently labelled probe and a specific surface component, using fluorescence polarisation (FP) for detection. The method analysed the change in polarisation of the emitted light between unbound and bound probes, with the observed polarisation in a mixture of the labelled probe and target being proportional to the fraction of bound probes. This property of FP allowed them to use it to quantify the amount of EVs in the solution. She explained the different strategies used to demonstrate the enhanced fluorescence polarisation in response to increasing EV concentration, quantified by NTA and the study of probe–target kinetics (O-15).

Finally, the last speaker, Enrico Ragni (IRCCS-Galeazzi Hospital, Milan, Italy) talked about the MSC secretome for regenerative medicine related to orthopaedic conditions. He characterised adipose-derived MSCs (ASCs) and EV-miRNAs and their modulation after high levels of IFNy preconditioning and mimicking OA. The penetration of ASC-EVs was evaluated in cartilage explants. Bioinformatics tools were used to predict the modulatory effect of the identified molecules on pathological cartilage and to follow and quantify the incorporation of fluorescent EVs into cartilage explants. The ASC secretome showed a strong propensity to modulate inflammatory and degenerative processes thanks to the detected presence of 50 cytokines/chemokines and more than 200 EV-miRNAs, and inflammatory preconditioning or OA-like conditions have been able to increase this ability. The ASC-secretome’s ability to stimulate healing and reduce inflammation allows it to be proposed as an ideal candidate for orthopaedic regenerative medicine [11] (O-16).

### 2.5. Session 5: “Advancing MSC Therapies in Veterinary Medicine: Present Challenges and Future Perspectives”

Laura Barrachina Porcar (University of Galway, Ireland) presented her work on MSCs and their immune properties. She explained that the use of allogeneic MSCs presented several advantages with respect to autologous cells as a possibility to have a ready-to-use product. Although cell-based products in the veterinary market are emerging, allogeneic therapy does not come without limitations. At first, MSCs, which were initially considered immune-privileged, can actually induce cellular and humoral immune responses. Based on these observations, she reported several human and animal studies with positive results after allogeneic MSC administration in the absence of adverse effects. A potential explanation for the mixed outcomes often seen was that MSCs can be recognised by the immune system (immunogenicity), but they can also regulate it (immunomodulation). She also explained the different strategies that could be designed to develop safer and more effective allogeneic therapies and how it could allow the creation of ‘haplo-banks’ of cells from donors. The understanding of the interactions between MSCs and the immune system could be a key to learn which factors we can manage, and how (O-17).

Following, the first oral selected communication was from Barbara Merlo (University of Bologna, Italy) who presented the results of a pilot study concerning the effect of GM18, an α4β1 integrin agonist, on the adhesion properties of equine adipose tissue and Wharton’s jelly derived MSCs and on their ability to adhere to GM18- incorporated poly L-lactic acid (PLLA) scaffolds. The use of biomaterials with integrin agonists promoted cell adhesion in tissue repair processes confirming the presence of GM18-containing PLLA scaffolds. In conclusion, GM18 affects equine MSCs adhesion ability with donor-related variability. These preliminary results suggested that MSCs from Wharton Jelly might be more suitable than MSCs from adipose tissue [12] (O-18).

The second selected oral communication was presented by Gabriele Scattini (University of Perugia, Italy) which was about migrasomes (MG), a particular kind of EVs released by MSCs of different species and tissue sources possibly related to their migratory activity. He reported that MGs were isolated by differential ultracentrifugation of MSC supernatant and were observed by Transmission Electron Microscopy (TEM) as different from other microparticles in size and morphology. Their biogenesis and morphologic features were described in detail compared to the EV characteristics, although he underlined that their different functions are still to be clarified (O-19).

### 2.6. Session 6: “MSCs: A Double-Edged Sword: Friend or Foe in Oncology?”

Roisin Dwyer (University of Galway, Ireland) opened the last session on the dual role of MSCs in oncology with a presentation of her research on the use of EVs isolated from MSCs as therapeutic delivery for cancer. MSC-derived EVs raised interest in their potential as tumour-targeted delivery vehicles for therapeutic agents. The recent work presented by Dwyer’s group showed the development of MSC-EVs enriched with a tumour suppressor microRNA for breast cancer therapy. Their objective was to develop an innovative approach based on MSC-EVs as a cancer treatment for patients with breast cancer who have limited treatment options. From her point of view, the use of more reflective pre-clinical models of the patient experience has to be developed to test new therapeutic approaches [13] (O-20).

Andrea Papait (Cattolica del Sacro Cuore University, Rome, Italy) demonstrated the MSC role in the tumour microenvironment and their complex interaction network in contact with anticancer agents. In his presentation, he discussed the use of MSCs considered as a double-edged sword: they can serve as a drug carrier while their immunomodulatory properties [14] can participate in tumour initiation, development and progression, and metastasis formation [15]. For him, in the era of immunotherapy, MSCs or their secretome could represent an adjuvant therapy in association with new drugs such as monoclonal antibodies aimed at re-educating the immune response (O-21).

Finally, the meeting ended with the selected presentation by Valentina Coccè (University of Milan, Italy). She presented data on the inhibitory effect of adipose tissue-derived MSCs uploaded by paclitaxel (PTX) on malignant pleural mesothelioma in vitro and in vivo models. In particular, she saw how adipose tissue (FAT) after micro fragmentation (MFAT) was able to exert a dose-dependent inhibition on the growth of the human mesothelioma cell line (MSTO-211H). While, in xenografted Balb/c-Nude mice obtained after subcutaneous injection with MSTO- 211H, she observed a reduction of tumour mass volume measurements after treatment with MFAT loaded with PTX similar to that of free PTX. This could be a new therapeutic approach for a difficult to treat tumour [16] (O-22).

## 3. Concluding Remarks

After each session, at least 30 min were dedicated to questions from the audience and replies from the speakers, making the discussion very open and engaging for all the presentations and also helping to draw out take-home messages. In particular:The first session about the source of funding for the research projects was very animated and of great interest for all the attendees.The “GISM—Next Generation” section, which primarily concentrates on improving general skills related to publishing, patent writing, fundraising, and knowledge transferring, provided valuable insights based on the latest technological advancements and opportunities presented by social networks.The third session, dedicated to new 3D culture systems, gave insight on the current 3D strategies that could aid in the creation of personalised therapy options with systems such as joints-on-chip or the MIVO system.The session on therapeutic applications of MSC-EVs in veterinary and human medicine provided that the mode of action in different diseases is essential for the MSC-EVs translation from the laboratory to the clinical settingThe fifth session showed the advancing MSC therapies in veterinary medicine present challenges and future perspectives not only in the veterinary field, but also for translation research. In fact, the MSC therapy in animals represents an excellent approach to have results describe an experimental investigation brochure dossier for an experimental clinical trial with secretome or MSCsThe last session was dedicated to how EVs can be used as drug carrier systems and how they interact with the tumour microenvironment, showing interesting new discoveries especially in breast and mesothelioma.

Figure 1 summarises the positive and negative aspects in mesenchymal stem cells utilisation and MSC secretome (microvesicles and exosomes), as raised from the six sections discussed here.

In general, the congress was a great success thanks to lively participation of researchers of all ages, demonstrating that the topics of the congress were very attractive in the scientific community. In fact, more than 100 researchers participated in the meeting and 51 posters were exposed and presented in a dedicated section of the congress.

Overall, the atmosphere was interactive and full of young researchers who felt at ease sharing their ideas and questions with senior mentors in all moments of the congress, including pleasant moments of relationship.

All the abstracts of oral and poster presentations, with written consent for the publication here.

## 4. Poster Award

During the 2022 GISM annual meeting, three “Young Investigator Awards” of 500 euros each were assigned. In order to be eligible, researchers had to (*a*) submit a spontaneous candidature; (*b*) be the first author of an accepted abstract; (*c*) be younger than 35 years on 20 October 2022; (*d*) be present at the Award Ceremony held during the “GISM—Next Generation” section. The three winners of the 2022 GISM annual meeting “Young Investigator Awards” were: Priscilla Berni (University of Parma, Italy), Elena Ceccotti (University of Turin, Italy), and Gianluca Cidonio (CLN2S, Fondazione Istituto Italiano di Tecnologia, Rome, Italy).

## 5. Abstracts

### 5.1. Oral Presentation



**O-01. BRIDGING THE GAP IN MEDICAL RESEARCH ON ADVANCED THERAPY MEDICINAL PRODUCTS: A BIOLOGICAL, REGULATORY AND MEDICAL EXPERIENCE**

**Graziella Pellegrini**
Centre for Regenerative Medicine “Stefano Ferrari”, University of Modena and Reggio Emilia, Modena, Italy
**Abstract**
Gene therapy, cell therapy, and tissue engineering have the potential to revolutionise the treatment of disease and injury. Attaining marketing authorisation for such advanced therapy medicinal products (ATMPs) requires a rigorous scientific evaluation by the European Medicines Agency—authorisation is only granted if the product can fulfil stringent requirements for quality, safety, and efficacy. However, many ATMPs are being provided to patients under alternative means such as “hospital exemption” schemes. Holoclar (ex vivo expanded autologous human corneal epithelial cells containing stem cells), a novel treatment for eye burns, is one of the few ATMPs to have been granted marketing authorisation and is the first to contain stem cells. This review highlights the differences in standards between an authorised and unauthorised medicinal product and specifically discusses how the manufacture of Holoclar had to be updated to achieve authorisation. The result is that patients will have access to a therapy that is manufactured to high commercial standards and is supported by robust clinical safety and efficacy data.




**O-02. BRIDGING THE FUNDING GAP IN MEDICAL RESEARCH IN ADVANCED THERAPY MEDICINAL PRODUCTS: TOWARDS A FAIR MEASUREMENT OF VALUE CREATION**

**Stefano Cosma, Daniela Pennetta, Francesca Guida**
University of Modena and Reggio Emilia, Modena, Italy
**Abstract**
The right to healthcare for every individual is crucial for social inclusion and sustainable development. Despite this, innovation and medical research face several difficulties in their path. The risks of failure in medical research are due to scientific and operational factors, but also to economic factors, particularly the lack or inadequacy of funding. In Finance, the prerequisite for a project to be funded is the possibility of correctly determining its overall value. This paper is the first step of a wider project that aims to measure the operational and financial needs of the research into ATMPs with a Capital budgeting approach in order to correctly determine the ability to generate (economic and social) value. The paper aims to explore the causes of failure of the various phases of medical research and if and how the current funding schemes or financial partners may affect them. The purpose is achieved with the support of the ClinicaTrial.gov database, which includes privately and publicly funded clinical studies conducted worldwide. As a preliminary study, we focused our attention on the Italian context, collecting information on interventional studies carried out by Italian research groups during the entire period of database coverage (since 1998 to 2021). We collected a total of 12,934 interventional studies, 344 of which we classified by keywords within ATMPs research. The analysis was also extended to a sample of 1137 European studies classified by keywords within ATMPs research. The study builds a complete, general picture of the current funding of clinical research, the current role of Finance and the types of involvement of the public sector and other non-profit partners. Through manual data reprocessing, we were able to highlight two important features of the studies in the sample. First, the information on their current status allowed us to pinpoint failed and succeeded projects, while also understanding failure causes. Second, information on their sponsor was re-elaborated in order to classify the studies according to the source of funding (e.g., industry, university, research centres, hospitals, and so on). Thanks to an intersection of this data, our work provides insights into how the current financial resource allocation may be correlated to the success or failure of clinical trials, with a focus on advanced therapies, revealing not only potential funding gaps but also implications for involved researchers, policymakers, and stakeholders.




**O-03. MY EXPERIENCE WITH MSCS IN ORTHOPAEDICS: PAST, PRESENT AND FUTURE**

**Nicholas Crippa Orlandi, Nicola Mondanelli, Stefano Giannotti**
Department of Orthopaedics and Traumatology, University of Siena, Italy
**Abstract**
Orthopaedic surgery can benefit a lot from the use of stem cells. Three case reports drawn from the experience of the authors’ Orthopaedic Clinic are illustrated to highlight the benefits of applying this technology. Drawing on the extensive experience gained within the authors’ Operating Unit, three cases regarding different body segments have been selected to prove the benefits deriving from the use of gelled preparations containing autologous MSC from bone marrow. A case of humeral shaft non-union, the management of an atypical proximal femur fracture in congenital hip dysplasia, and a case of rupture of the patellar tendon and consequent reconstruction of the extensor apparatus with a cadaveric transplant. The experimental study, whose three cornerstones are mesenchymal stem cells (hMSCs), scaffolds, and growth factors, aims to develop new tissue engineering strategies to be applied in orthopaedic surgery. In the first part of the work, the focus was on the optimisation of the isolation and expansion protocol of the hMSCs, taken from the bone marrow and deposited in the Biobank present inside the laboratory which is part of the Network Telethon (TNGB), evaluating the replacement of the FBS (fetal bovine serum) with autologous PRP (platelet rich plasma). Then, the focus shifted to the characterisation of cells seeded on two types of scaffolds: bovine bone matrix (SmartBone) and lyophilized acellular dermis. The final goal of the work is to develop strategies to increase the osteogenic properties of the support by adding growth factors that enhance cell differentiation on the scaffold. Our technique of applying stem cells to cases of complex orthopaedic surgery has shown excellent results both in a clinical functional objective and subjective evaluations and in radiographic evaluations. The experimental study, currently underway, is providing excellent results. The data suggest PRP as a valid alternative to FBS, it supports the expansion of mesenchymal stem cells without compromising their capacity since cells loaded on SmartBone differentiate into osteoblasts and produce collagen. In our opinion, MCSs are and will become more and more a valid tool to provide the surgeon with important help in cases of great complexity and, therefore, to obtain the tailored care that every patient needs and deserves.




**O-04. ORAL CAVITY MSC PHENOTYPING FOR REGENERATIVE MEDICINE APPLICATIONS**

**Ilaria Roato, Tullio Genova, Beatrice Masante, Giacomo Baima, Alessandro Mosca Balma, Federico Mussano**
University of Turin, Italy
**Abstract**
The oral cavity contains multiple sites of MSCs, which have been studied as promising candidates for tissue regeneration in the dental/maxillofacial and neuroregenerative field due to the neural crest derivation of most of them. Adult dental pulp stem cells (DPSCs) are more indicated for the pulp–dentin complex regeneration while proved to be less osteogenic compared to buccal fat pad stem cells (BFPSCs). The analysis of the immunophenotype of in vitro expanded stem cells from human exfoliated teeth (SHED), DPSCs, and periodontal ligament stem cells (PDLSCs) showed a comparable expression of MSC markers among them and a higher percentage of endothelial precursor subset in SHED and DPSCs than in PDLSCs. Owing to their peculiar origin, MSCs isolated from the oral cavity might be more effective than adipose-derived stem cells (ASCs) for the treatment of dental defects. Indeed, even though MSCs retrieved from different tissues show comparable immunophenotype and multilineage differentiation ability, their capability to differentiate into a specific tissue depends also on the different anatomical origins. Moreover, often, these cells may be harvested without any burden or additional discomfort for the patient who undergoes a tooth extraction for orthodontic indications.




**O-05. A NEW STRATEGY TO MONITOR MESENCHYMAL STEM CELLS AND EXTRACELLULAR VESICLES IN ADVANCED THERAPIES APPLICATION**

**Marta Nardini ^1^, Maria Elisabetta Federica Palamà ^2^, Vipul Gujrati ^3^, Vasilis Ntziachristos ^3^, Mary Murphy ^4^, Niamh Duffy ^4^, Ignacio de Miguel ^5^, Martin Leahy ^6^, Chiara Gentili ^2^ and Maddalena Mastrogiacomo ^1^**




^1^ Department of Internal Medicine (DIMI), University of Genoa, Italy^2^ Department of Experimental Medicine (DIMES) University of Genova, Italy^3^ Chair of Biological Imaging at the Central Institute for Translational Cancer Research (TranslaTUM), School of Medicine, Technical University of Munich, Germany^4^ Regenerative Medicine Institute, School of Medicine, University of Galway, Ireland^5^ Knowledge & Technology Transfer department. Institut de Ciències Fotòniques (ICFO), The Barcelona Institute of Science and Technology, Barcelona, Spain^6^ National University of Ireland, National Biophotonics and Imaging Platform, School of Physics, Tissue Optics and Microcirculation Imaging Group, Galway, Ireland




**Objective**
An important challenge in regenerative medicine to the regulatory approval and widespread clinical acceptance of cell therapies is surrounding the behaviour of cells after transplant. The distribution, engraftment, viability, and activity of transplanted cells are unclear. This reduces confidence in the safety of cell therapies and makes it difficult to understand the mechanism of action for developing potency assays that will be required for advanced clinical testing and the approval of cell therapy. When Extracellular Vesicles (EVs) are used as therapeutic agents, similar issues arise. Optoacoustic imaging (OAI) is a rapidly developing technology with world-leading capabilities in functional imaging that is uniquely informative and lower cost, convenient, and rapid. Gold-Nanostars (GNs) are suited for use as a contrast medium for optoacoustic imaging that generates a very strong photo-thermal signal in response to light of longer wavelengths. The combination of OAI and GNs offers a class-leading imaging solution for cell therapies in regenerative medicine. In this work after a characterisation in vitro of the complexes, GNs-labelled Mesenchymal Stem Cells (MSC) and EVs were studied for their distribution in different organs after a systemic injection by visualisation with Multispectral Optoacoustic Tomography (MSOT) and tested for the in vivo toxicity.
**Materials and Methods**
**MSCs** were isolated from human bone marrow, expanded in Platelet Lysates (PL), and derived from the same cell EVs. The MSC and EVs were labelled with GNs and by in vitro tests with the right concentration of particles, in terms of capability to be visualised by MSOT and no cytotoxicity, was defined. Labelled and unlabelled complexes were injected into the caudal vein of male and female Nu/Nu mice. Blood tests and histological analysis of the organs were performed at different time points to test the in vivo toxicity.
**Results**
No alterations were observed in terms of biochemical and blood markers. MSOT analysis showed that after one day, GNs reached all organs and remained for all monitoring periods without any alteration as revealed by histological analyses. In particular, MSOT analysis showed that GNs cannot overcome Blood–Brain Barrier (BBB).
**Conclusions**
The results showed that the GNs can reach the various organs and that they persist there for some time, but although they persist, they cannot overcome BBB and does not lead to any type of alteration. This approach holds promise for tracking cell distribution in the tissue repair process.




**O-06. SCIENTIFIC ARTICLES: HOW TO CHOOSE THE RIGHT JOURNAL TAKING ADVANTAGE OF SOME OPPORTUNITIES**

**Filippo Piccinini ^1,2^**




^1^ IRCCS Istituto Romagnolo per lo Studio dei Tumori (IRST) ‘‘Dino Amadori”, 47014 Meldola, Forlì-Cesena, Italy^2^ Department of Medical and Surgical Sciences (DIMEC), University of Bologna, 40136 Bologna, Italy




**Abstract**
“Scientific Articles” are not the final goal but a necessary intermediate step for worldwide progress. “Scientific journals” are the right place to communicate discoveries to society. “Editors and Reviewers” are important people for evaluating the results. “Impact Factor” is one of the most common words in the life of a researcher. These terms are just a few of many behind the real important things in research including “Hypotheses and Results”. However, a large part of the working life of a researcher is spent writing scientific articles and finding the right journals for them. In this presentation, we will discuss how to speed up the publication process, define the best journal, and increase the chances of publication by exploiting opportunities that technology and social networks today provide us.




**O-07. TECHNOLOGY TRANSFER AND INTELLECTUAL PROPERTY: LET’S TALK TO AN EXPERT**

**Paola Bagnoli**
IRCCS Ospedale Galeazzi Sant’Ambrogio of Milan, Italy
**Abstract**
Technology transfer in the Life Science field is a demanding but very stimulating challenge for both researchers and technology transfer experts. The sector has peculiarities that differentiate it from most other technological fields and make the exploitation of research results even more challenging.The results obtained by universities and research hospitals, while promising, are often premature and require a long development path to be commercialised and brought to the clinic. It is well known that very long research and development times and high costs are required to bring a medical device or drug to the market. This gap between proposals from the research world and needs from industry is well known by the technology transfer experts and often discourages the researchers who do not see their research reaching the patient, although a strong clinical need is well identified. Many factors are crucial to fill this gap, not just the need for dedicated funding. The design according to the correct principles is fundamental to allow the industrialisation of the prototype, the scale up of the processes, and the subsequent certification phases. Last but not least, the protection of industrial property (IP) is crucial. Indeed, a company would be unlikely to invest millions of euros in a new life science project without the certainty of exclusivity on the results that would guarantee a competitive advantage over its competitors. The patent is the main tool to obtain this exclusivity, together with the copyright, the design and the trade secret. The proper protection of the research results is an essential step in the technology transfer process. Researchers can have support from the Technology Transfer Offices (TTO) of their institutions, which have the fundamental role of scouting the results, assisting researchers in protecting IP and then activating virtuous paths for the development of the Technology Readiness Level (TRL) of the projects. These paths have to be walked together with several stakeholders such as corporates, start-up incubators, business angels, venture capitalists, etc. The activation of co-development processes with companies and grants dedicated to the development of the Proof of Concept of the patented technologies are key steps in making the project attractive for a company that can licence it or in creating a startup to develop and commercialise the new technology.




**O-08. RECAPITULATING MUSCULOSKELETAL TISSUE COMPLEXITY THROUGH MICROFLUIDIC AND BIO-FABRICATED 3D MODELS**

**Matteo Moretti**
IRCCS Istituto Ortopedico Galeazzi of Milan, Italy
**Abstract**
The musculoskeletal system is composed of different tissues and organs, i.e., bones, muscles, tendons, and joints. Several pathologies can affect its homeostasis including traumatic injuries and inflammatory diseases such as osteoarthritis (OA). In this context, mesenchymal stem cells have been considered either as building blocks to fabricate biological substitutes for damaged musculoskeletal tissues due to their differentiation potential towards bone and cartilage or as a possible therapy due to their anti-inflammatory properties. On the other hand, 3D in vitro models are increasingly being considered a powerful tool to investigate pathological mechanisms and therapeutic efficacy, allowing to overcome the excessive simplification of standard in vitro models and species-specific differences of animal models. Thus, we developed 3D in vitro models of joints to test the potential of mesenchymal cells as anti-OA therapy and of the muscle–tendon–bone junction, based on differentiated mesenchymal cells. We generated a microfluidic joint-on-a-chip model including cartilage, synovial membrane and synovial fluid, based on patient-matched synovial fluid and cells embedded in gels mimicking the composition of cartilage ECM. The injection of OA synovial fluid allowed for the reproducing of the main hallmarks of early OA as the production of degradative enzymes and an increase in inflammatory cytokines. Furthermore, we were able to detect the effects of mesenchymal cell injection, measuring MMPs and inflammatory cytokine release, on a patient basis, opening new possibilities for the establishment of personalised treatments. We also fabricated a 3D bioprinted muscle–tendon–bone model embedded in a microfluidic chip, allowing the fabrication and culture of three different connective tissues in their specific culture media. Computational simulations were performed to design the chip, keeping the culture media separated during perfusion. The chip also allowed to apply compression to the bone and stretch to muscle and tendon in physiological and pathological ranges, simulating traumatic conditions. The 3D bioprinting procedure resulted in three separated cell compartments, maintaining a good shape fidelity and high cell viability. In conclusion, biofabricated 3D in vitro models could help foster the application of mesenchymal cells as anti-inflammatory therapy and could benefit from their potential as starting material for the reconstruction of musculoskeletal tissue units.




**O-09. EXTRACELLULAR VESICLE-MEDIATED COMMUNICATION ROUTES IN 3D TUMOUR MODELS**

**Maria Harmati ^1^, Akos Diosdi ^1,2,3^, Ede Migh ^1^, Gabriella Dobra ^1,4^, Timea Böröczky ^1,4^, Edina Gyukity-Sebestyen ^1^, Matyas Bukva ^1,4^, Sandor Körmöndi ^5^, Peter Horvath ^1,3,6^, Krisztina Buzas ^1,7^**




^1^ Biological Research Centre, Eötvös Lorand Research Network, Szeged, Hungary^2^ Doctoral School of Biology, University of Szeged, Hungary^3^ Single-Cell Technologies Ltd., Szeged, Hungary^4^ Doctoral School of Interdisciplinary Medicine, University of Szeged, Hungary^5^ Department of Traumatology, University of Szeged, Hungary^6^ Institute for Molecular Medicine Finland (FIMM), University of Helsinki, Finland^7^ Department of Immunology, University of Szeged, Hungary




**Objective**
The evolutionary process of solid tumours highly relies on the extracellular vesicle (EV)-mediated crosstalk between malignant cells and stromal cells in the tumour microenvironment (TME). In this study, we aimed to establish a multicellular three-dimensional (3D) tumour model system for tracking the EV communication network of different tumour tissues under physiological conditions and cytostatic treatments.
**Materials and Methods**
Human ductal carcinoma, melanoma, and osteosarcoma models were established via co-culturing the respective tumour cell line (T-47D/A375/MG-63) with MRC-5 fibroblasts and EA.hy926 endothelial cells on flat- or U-bottom plates after staining with CellTracker dyes (Orange CMTMR, Deep Red, Green CMFDA). To mimic chemotherapeutic stress, low dose doxorubicin was used and the 2D and 3D cultures were imaged daily by a PerkinElmer Operetta High Content Screening System and a Leica SP8 Digital LightSheet microscope, respectively.
**Results**
Preliminary experiments showed that CellTracker dyes can be used for in-cell labelling of EVs, allowing the quantitative monitoring of EV crosstalk, i.e., EV routes between each cell type and in both directions. The three types of tumour models showed differences in their 3D structure, EV crosstalk activity, and drug-induced effects as well. We could observe distinct temporal kinetics in the development of the EV communication network in 2D and 3D, also priorities of the investigated EV routes varied between the two co-culture systems.
**Conclusions**
The developed 3D model system is suitable for live tracking of EV crosstalk in the TME, which enables the comparison of the primary EV communication routes in different tumour types and drug treatments. Further data will help (i) to identify potential targets of EV-blocking therapies, which may increase the efficacy of chemotherapies, and (ii) to predict the drug-induced changes of the communication activity in different tumour tissues.




**O-10. DEMONSTRATING METHOD SUITABILITY FOR A RAPID MICROBIAL DETECTION METHOD APPLICABLE TO BIOLOGIC PRODUCTS**

**Lucia Ceresa, Senior Technology and Market Development Manager**
Microbial Solutions, Charles River, Italy
**Abstract**
New and emerging cell therapies and medicinal products present new challenges in the assurance of quality and safety in terms of end-product testing. While traditional pharmaceutical drug products have long-established standards for sterility assurance, these established processes are not optimised for cell therapies. For example, the Compendial Sterility Test requires more than fourteen days of incubation for a reliable result, making it a rate limiter in the distribution of therapies to patients. Rapid Microbial Methods (RMM’s) offer reliable alternatives to ageing microbiological methods to solve the problem of reducing cycle time in cell therapy manufacture. ATP Bioluminescence is a matured technology that is used in the quality testing of various product samples in different industries. Detection of microbial ATP using the luciferin-luciferase reaction allows for the detection of microbes before they can be cultured to visual detection levels on microbial media. Until recently, ATP bioluminescence was not a viable contamination detection option for cell-based products because these samples also contain cellular ATP. The established ATP Bioluminescence platform, Celsis^®^, was further developed to address this limitation. A sample cell lysing procedure allows for the extraction and, more important, the depletion of “non-microbial-ATP”, while leaving microbial ATP intact. A case study on tests performed on different cell lines demonstrate the detection of the “slow growing” *C. acnes* as well as a wide panel of other typical and critical microbial species. Studies performed using the Celsis^®^ platform and the Celsis Adapt™ complimentary technology demonstrated the successful depletion of cellular ATP from samples, while also allowing fast detection of microbial presence contamination for superior microbiological contamination control.




**O-11. A NOVEL ORGAN ON CHIP PLATFORM FOR CULTURING 3D HUMAN TISSUES UNDER PHYSIOLOGICAL FLOW-CONDITIONS FOR MORE PREDICTIVE DRUG TESTING AND HUMAN DISEASE MODELLING**

**Silvia Scaglione ^1,2^, Monica Marzagalli ^1^**




^1^ React4life srl, Genoa, Italy^2^ CNR-IEIIT Institute, National Research Council of Italy, Genoa, Italy




**Objectives**
The human disease modelling for basic research and drug testing purposes is currently carried out through 2D cell culture in static conditions and in vivo xenografts or genetically engineered animal models, but predictability, reliability, and complete immune compatibility remain important challenges. For this aim, novel 3D, fully humanised in vitro tissue models have been recently investigated by adopting emerging technologies such as bioprinting and microphysiological systems, also named organ on chips. A novel Multi-In Vitro Organ (MIVO) organ on a chip platform has been recently developed to culture 3D clinically relevant size tissues under proper physiological culture conditions.
**Material and Methods**
Biologically relevant cancer samples (up to 5 mm), and patient biopsies of scaffold-based tissues have been cultured within the MIVO chamber, while either testing molecules or human immune cells (e.g., Natural Killer cells, NK) can circulate in the OOC mimicking the blood capillary flow. The cell proliferation, viability, and migration within 3D matrixes were investigated in such dynamic cell culture conditions. When systemic drug administration was simulated within the OOC, the anticancer drug efficacy was tested and compared to the animal model. When NK cells were placed in circulation, their extravasation through a permeable barrier resembling the vascular barrier and infiltration within the cancer tissue were analysed.
**Results**
A human 3D ovarian model was developed and treated with Cisplatin in static conditions within MIVO and in the xenograft model. Similar tumour regression was observed in MIVO and in mice, while the static culture displayed an unpredicted chemoresistance, due to unreliable drug diffusion. A human 3D neuroblastoma cancer model with proper immunophenotype was optimised to develop a complex tumour/immune cell co-culture as a paradigm of an immune-oncology screening platform. Importantly, a tumour-specific NK cell extravasation was observed with a tumour-specific NK cell infiltration within 3D tumour tissue and cancer cells apoptosis induction.
**Conclusions**
We generated a relevant human disease mode, through the adoption of the MIVO device, that can be efficiently employed as a drug screening platform, but also for better investigating crosstalk among immune cells and other healthy/pathological tissues.




**O-12. TRANSLATIONAL LAB-TO-CLINIC HURDLES IN THERAPY WITH MSC-EVS**

**Stefania Bruno**
Department of Medical Sciences, University of Turin, Italy
**Abstract**
Extracellular vesicles (EVs) derived from mesenchymal stromal cells (MSCs) have been demonstrated to have therapeutic effects in pre-clinical models of different diseases. EVs derived from various types of MSCs have pro-regenerative capabilities comparable to cells of origin and are considered promising tools for the treatment of a variety of acute and chronic pathologies. To this end, the characterisation (size, phenotype, molecular content, etc.) of EVs and the evaluation of their biological effects are areas of intense investigation. To successfully translate EV research along the path from the laboratory to the patients, strategies should be designed to reach the aim of clinical testing of MSC-EVs safety and efficacy. Currently, there are emerging strategies for manufacturing cells and EVs and quality controls for practicability of clinical testing of the EVs, mainly related to non-industrial processes. Moreover, the identification of the mode of action in the therapeutic approach in the different diseases remains a major challenge to the translation path of EVs from the laboratory toward the clinical setting.




**O-13. CELECTOR^®^, AN INSTRUMENT FOR QUALITY CONTROL AND STANDARDISATION OF ATMPS**

**Silvia Zia, Pasquale Marrazzo, Laura Bonsi, Francesco Alviano, Barbara Roda, Andrea Zattoni, Pierluigi Reschiglian**
Stem Sel srl, Bologna, Italy
**Abstract**
Cell therapy represents innovative medical approaches in many clinical fields such as osteo-articular reconstructive surgery, tissue engineering, and cancer. Advanced therapy medicinal products (ATMPs) are cell and tissue products that are considered new types of drugs. Mesenchymal stromal stem cells (MSCs) are the most promising candidates in current clinical trials. Besides their regenerative properties, hMSCs have also shown high immunomodulatory potential. The ATMP products should follow requirements including sterility, identity, purity, viability, potency, and reproducibility. Because stem cells are a heterogenous population that can differ depending on the origin and manipulation of the starting material, the identity/purity of the final cell population and its yield is still a critical issue which can limit clinical application of MSC-based products. New approaches for the standardisation of cell-based protocols may allow for the development of new high-efficiency drug systems. To improve MSC characterisation, novel label-free functional tests, evaluating the biophysical properties of the cells, will be advantageous for their cell profiling, population sorting, and quality control. In this work, we present a new technology, Celector^®^, for the quality control of the cell population. Celector^®^ has been shown to be able to tag-less analyse, discriminate, and separate a wide size range of cells based on their physical characteristics, with high resolution and throughput and with total maintenance of native properties. The separation is obtained in a short time (around 15 min) in a rectangular shape capillary device, through to the combined action of gravity, acting perpendicularly to the flow and opposing lift forces that depend on the morphological features of the sample. A micro-camera is connected as a detector and specifically designed software was developed for image acquisition, post-processing and data analysis, and the fingerprint of the biological sample. Celector^®^ is able to highlight physical differences that can be correlated to cell viability and regenerative potential. In addition, cells with stable and reproducible doubling time analysis can be collected and used as standardised systems for the development of high-quality clinical protocols.




**O-14. HUMAN LIVER STEM CELL-DERIVED EXTRACELLULAR VESICLES INTERFERE WITH THE DEVELOPMENT OF CHRONIC KIDNEY DISEASE IN AN IN VIVO EXPERIMENTAL MODEL OF RENAL ISCHEMIA AND REPERFUSION INJURY**

**Elena Ceccotti ^1^, Massimo Cedrino ^2^, Giulia Chiabotto ^1^, Cristina Grange ^1,3^, Samuela De Rosa ^1^, Giovanni Camussi ^1,3^, Stefania Bruno ^1,3^**




^1^ Department of Medical Sciences, University of Turin, Italy^2^ Unicyte AG, Obendorf, Switzerland^3^ Molecular Biotechnology Center, University of Turin, Italy




**Objective**
Renal ischemia reperfusion injury (IRI) is the major cause of acute kidney injury (AKI), and it increases the risk of progression to chronic kidney disease (CKD). Human liver stem cells (HLSCs) are a mesenchymal stromal cell (MSC)-like population isolated from adult liver biopsy. HLSCs share with MSCs the same phenotype, gene expression profile, and differentiation capabilities. As shown in previous studies, HLSCs improved the recovery in different experimental models of liver and kidney injury. HLSC-derived extracellular vesicles (HLSC-EVs) have been studied as vehicles for transferring active biological materials both in vitro and in vivo. In this study, we set up an in vivo murine model of IRI-AKI, which subsequently developed into IRI-CKD and we investigated the potential therapeutic effect of HLSC-EVs.
**Materials and Methods**
EVs were purified from HLSC supernatant by ultracentrifugation. They were analysed through flow cytometry and Western Blotting to detect the expression of the main mesenchymal and EV surface markers. Transmission electron microscopy was performed to analyse EV size and morphology. Male BALB-c mice were subjected to 30 min ischemia followed by reperfusion. HLSC-EVs were intravenously administered immediately after the surgery and three days after. To evaluate AKI, mice were sacrificed two and three days after the surgery; while to assess the development of CKD, mice were sacrificed two months after. The histological analyses were performed on tissue sections stained with hematoxylin and eosin and Masson’s trichrome for collagen detection. Expression of specific markers of fibrosis development (alpha-Smooth Muscle Actin (alpha-SMA), collagen I and transforming growth factor-beta) and inflammation (interleukin-1 beta, interleukin-6, and tumour necrosis factor-alpha) were evaluated at mRNA and protein levels.
**Results**
In AKI mice, the EV treatment-attenuated kidney damage by reducing tubular necrosis and increasing tubular cell proliferation. We also noticed the downregulation of the expression levels of fibrosis-related genes. In CKD mice, EVs effectively reduced the development of interstitial fibrosis at the histological level and reduced the expression levels of pro-fibrotic and pro-inflammatory genes.
**Conclusions**
The administration of HLSC-EVs immediately after renal IRI protects the kidney from AKI development and interferes with the development of subsequent CKD.




**O-15. FLUORESCENCE POLARISATION-BASED EXTRACELLULAR VESICLES QUANTIFICATION: APTAMER APPROACH**

**Tarlan Eslami Arshaghi ^1^, Jerry Clifford ^2^, Stephanie J Davies ^2^, Frank Barry ^1^**




^1^ Regenerative Medicine Institute, University of Galway, Ireland^2^ Valitacell Ltd., Dublin, Ireland




**Objective**
Extracellular vesicles (EVs) have attracted wide interest in recent years due to their potential applications in regenerative medicine, as biomarkers for disease diagnosis, and their role in cell–cell interactions. However, EV isolation, quantification, and characterisation remain challenging in terms of purity and specificity as well as time- and cost-effectiveness. This work aims to develop a novel and high-throughput EV quantification tool based on the interaction between a fluorescently labelled probe and a specific EV surface component, using fluorescence polarisation (FP) for detection. The method analyses the change in the polarisation of emitted light, between unbound, and bound probes, with the observed polarisation in a mixture of the labelled probe and target being proportional to the fraction of the bound probe. This property of FP allows us to use it to quantify the amount of EVs in the solution.
**Materials and Methods**
Two distinct strategies have been investigated, with probes targeting (i) specific EV surface markers (Tetraspanins, e.g., CD63) for EV sub-population quantification or (ii) the EV phospholipid bilayer membrane for total EV quantification. Commercially available fluorescently labelled CD63 binding aptamers and proteins and lipophilic dyes have been evaluated. EVs derived from HEK and MSC cultures have been purchased or isolated through PEG-precipitation and ultracentrifugation, and particles quantified through Nano Tracking Analysis (NTA) or Imaging Flow Cytometry. Each probe candidate was incubated with EVs, and FP was measured over time using a Spark Cyto plate reader.
**Results**
Tetraspanin specific (Anti-CD63) aptamer probes and lipophilic dyes have demonstrated increased fluorescence polarisation in response to increasing EV concentration quantified by NTA.
**Conclusions**
This initial proof of concept supports the use of FP as a high throughput EV detection and quantification method, with the ability to provide both total particle and CD63^+ve^ particle numbers. Further investigation is required to demonstrate the specific binding of each probe to its target to benchmark the FP assay against currently available methods. For this purpose, Bio-Layer Interferometry (BLI) assay is under examination to study the probe–target kinetic.




**O-16. MESENCHYMAL STROMAL CELL SECRETOME FOR REGENERATIVE MEDICINE: MODULATION OF SOLUBLE FACTORS AND EXTRACELLULAR VESICLES EMBEDDED-miRNAs BY DIFFERENT CULTURING CONDITIONS FOR JOINT DISEASES**

**Enrico Ragni ^1^, Paola De Luca ^1^, Carlotta Perucca Orfei ^1^, Alessandra Colombini ^1^, Marco Viganò ^1^, Francesca Libonati ^1^, Stefania Cicolari ^1^, Leonardo Mortati ^2^, Laura de Girolamo ^1^**




^1^ IRCCS Istituto Ortopedico Galeazzi, Laboratorio di Biotecnologie Applicate all’Ortopedia, Milan, Italy^2^ INRIM, Istituto Nazionale di Ricerca Metrologica, Turin, Italy




**Objective**
In regenerative medicine approaches related to orthopaedic conditions, mesenchymal stromal cells (MSCs) showed positive outcomes due to the secretion of therapeutic factors, both free and conveyed within extracellular vesicles (EVs), collectively termed secretome. MSC-derived factors may be modulated by both culturing and in vivo conditions. Nevertheless, a homogenous and comprehensive fingerprint in the frame of orthopaedic applications is missing. The aim of this work was to characterise adipose-derived MSC, (ASC)-secreted factors, and EV-miRNAs and their modulation after high levels of IFNγ preconditioning, as proposed for clinical-grade production of secretome with improved potential or low levels inflammatory conditions, mimicking osteoarthritis (OA) and synovial fluid (SF). In addition, ASC-EVs penetration in cartilage explants was scored.
**Material and Methods**
ASCs were cultured with and without IFNγ (1 ng/mL) or TNFα (5 pg/mL) + IL1β (10 pg/mL) + IFNγ (40 pg/mL) mimicking OA-SF. First, 200 secreted factors were assayed by ELISA. Second, 754 miRNAs were searched by qRT-PCR in ultracentrifuge-purified EVs. Bioinformatics tools were used to predict the modulatory effect of identified molecules on pathologic cartilage and synovial macrophages. Time-lapse coherent anti-Stokes Raman scattering, second harmonic generation, and two-photon excited fluorescence were used to follow and quantify fluorescent EVs incorporation into cartilage explants.
**Results**
Data showed that more than 50 cytokines/chemokines and more than 200 EV-miRNAs could be identified. The vast majority of molecules are involved in extracellular matrix remodelling and homeostasis of inflammatory cells. Inflammatory priming and synovial fluid-like conditions were able to further increase the capacity of the secretome to stimulate healing and inflammation reduction. Eventually, EV penetration was monitored as a fast process, starting in a few minutes, and reaching 30–40 μm depth after 5 h and plateau at 16 h in both cells and matrix of the cartilage explants.
**Conclusions**
Due to the portfolio of soluble factors and EV-miRNAs, the ASC secretome showed a strong propensity to modulate inflammatory and degenerative processes. Inflammatory preconditioning or OA-like conditions were able to increase this ability. Eventually, microscopy data supported the capacity of MSC-EVs to influence the chondrocytes embedded in their native ECM by active interaction and eventual therapeutic cargo release.




**O-17. MESENCHYMAL STEM CELLS AND IMMUNE PROPERTIES: WHAT HAVE WE LEARNED FOR THEIR ALLOGENEIC APPLICATION**

**Laura Barrachina Porcar**
Regenerative Medicine Institute (Remedi), University of Galway, Ireland
**Abstract**
Allogeneic MSCs present several advantages and the first allogeneic cell-based products in the veterinary market are emerging. Quality cells can be banked to be ready to use, avoiding the delay inherent to expanding autologous cells, which in addition may be unsuitable due to the patient’s age, genetic, or metabolic disorders. However, allogeneic therapy does not come without limitations. At first, MSCs were considered immune-privileged, but currently, we know that allogeneic MSCs can induce cellular and humoral immune responses. The immune targeting of MSCs can impact their properties and lead to adverse events, thus affecting their therapeutic efficacy and safety. Nevertheless, several human and animal studies report positive results after allogeneic MSC administration. A potential explanation for the mixed outcomes often seen is that MSCs can be immunogenic (i.e., able to raise an immune response) but they also are immunomodulatory. The less immunogenic and the more regulatory MSCs are, the better chances they have to elicit their therapeutic actions. However, several factors can affect the balance between these two immune properties. MSC immune recognition is mainly mediated by the major histocompatibility complex (MHC), which is variably expressed depending on the donor, source, or in vitro/in vivo conditions. In addition, the MHC matching between donor and recipient determines the immune recognition as in organ transplants. Although MSCs can be recognised by the immune system, they can also regulate it. Factors such as inflammation or differentiation can affect both MSC immunogenicity and immunomodulation, so it is key to learning how different conditions affect this balance. Based on this knowledge, different strategies can be designed to develop safer and more effective allogeneic therapies. The selection of the donor based on its MHC haplotype and expression pattern would allow the creation of ‘haplo-banks’ of cells from donors homozygous for the most common MHC haplotypes, as it has been proposed for human iPSCs. Other strategies aim at either decreasing the immunogenicity or increasing the immunomodulation of MSCs. For example, some growth factors can decrease the MHC expression and pro-inflammatory cytokines can increase MSC immune suppression. Understanding the interactions between MSCs and the immune system is key to learning which factors we can manage and how in order to enhance the clinical safety and effectiveness of allogeneic cell therapies.




**O-18. PEPTIDE MEDIATED ADHESION TO BETA-LACTAM RING OF EQUINE MESENCHYMAL STROMAL CELLS: A PILOT STUDY**

**Barbara Merlo ^1,2^, Vito Antonio Baldassarro ^1,2,3^, Alessandra Flagelli ^2^, Romina Marcoccia ^2^, Valentina Giraldi ^2,4^, Maria Letizia Focarete ^2,4^, Daria Giacomini ^2,4^, Eleonora Iacono ^1,2^**




^1^ Department of Veterinary Medical Sciences, University of Bologna, Italy^2^ Interdepartmental Center for Industrial Research in Health Sciences and Technologies, University of Bologna, Italy^3^ IRET Foundation, Ozzano Emilia, Italy^4^ Department of Chemistry “Giacomo Ciamician” and INSTM UdR of Bologna, University of Bologna, Italy




**Objective**
The use of biomaterials with integrin agonists could promote cell adhesion in tissue repair processes. Studies on the use of scaffolds with integrin agonists based on β-lactams in equines have never been reported. The aim of this study was to analyse the effect of GM18 (α4β1 integrin agonist) on cell adhesion of equine adipose tissue (AT) and Wharton’s jelly (WJ) mesenchymal stromal cells (MSCs) and to investigate the cell adhesion to GM18-incorporated poly L-lactic acid (PLLA) scaffolds.
**Materials and Methods**
Scaffold was fabricated using a homemade electrospinning apparatus consisting of a high-voltage power supply, a syringe pump, a glass syringe containing the PLLA solution, and connected to a stainless steel blunt-ended needle through a PTFE tube. Adhesion assays were performed after culturing AT and WJMSCs with coating or soluble GM18. Gene expression of the target integrins was evaluated. Cell adhesion on GM18 containing PLLA scaffolds after 20 min and 24 h coincubation was assessed using the two samples of AT and WJMCSs more sensitive to GM18. A cell-based high content screening was used for analyses. For statistical analyses, 2- or 1- Way ANOVA was used. Results were considered significant for *p* < 0.05.
**Results**
Soluble GM18 affects the adhesion of equine AT and WJMSCs, even if its effect is variable between donors. For ATMSCs, the presence of GM18 did not affect the adhesion to the PLLA scaffold, mainly due to a high variability detected at a concentration of 10%. Moreover, cultures seeded on the PLLA control scaffold cannot increase cell number after 24 h, reflecting an impairment in the early proliferation. However, the presence of GM18 in all the analysed concentrations induces an increase in cell number after 24 h (5%, *p* = 0.0170; 10%, *p* = 0.0144; 15%, *p* = 0.0395). For WJMSCs, the cell adhesion was affected after 24 h (*p* = 0.0427), drastically increasing due to the highest concentration of GM18 (*p* = 0.0368). The same concentration is also the only condition producing an increase in cell number after 24 h (*p* = 0.0021).
**Conclusions**
In conclusion, the α4β1 integrin agonist GM18 affects equine AT and WJMSCs adhesion ability with a donor-related variability. These preliminary results represent a first step in the study of equine MSCs adhesion to PLLA scaffolds containing GM18, suggesting that WJMSCs might be more suitable than ATMSCs. However, the results need to be confirmed by increasing the number of samples before drawing definite conclusions.




**O-19. MESENCHYMAL STROMAL CELL-DERIVED MIGRASOMES: A MORPHOLOGICAL STUDY**

**Gabriele Scattini, Luisa Pascucci**
University of Perugia, Department of Veterinary Medicine, Perugia, Italy
**Objective**
Cell migration is fundamental in numerous physiological and pathological processes, including tissue regeneration. Migrasomes (MG) are large structures growing on the cell surface and on the tip of nanotubes emerging from cell body during migration. MG represents a distinct type of extracellular vesicles (EVs) and are significantly different from other microparticles in size, morphology, and function. In this study, a morphological analysis of MG generated by Mesenchymal Stromal Cells (MSC) of different species was performed.
**Material and Methods**
EVs were isolated by differential ultracentrifugation. Briefly, the conditioned medium was centrifuged at 300× *g* to pellet cells. The supernatant was centrifuged at 2000× *g* (2 K EVs fraction), at 10,000× *g* (10 K EVs fraction), and finally at 100,000× *g* (100 K EVs fraction). EVs suspensions were placed on formvar-coated copper grids, contrasted with 2% uranyl acetate, and observed under a Philips EM 208 electron microscope (TEM) equipped with a digital camera. MSC monolayers were also fixed with 2.5% glutaraldehyde and 1% osmium tetroxide, dehydrated, and resin embedded. 80 nm thick sections were contrasted with 2% uranyl acetate and observed at TEM.
**Results**
Ultrastructural analysis revealed that MSC produces a previously unrecognised kind of vesicles, referred to as “migrasomes” that originate from the cell surface. The biogenesis and morphologic features of these vesicles are completely different from typical EVs. In fact, they are very large in diameter (500–2000 nm) and contain a variable number of luminal vesicles. MG were also detected in the 2 k fraction obtained by supernatant centrifugation.
**Conclusions**
Migrasomes are a special kind of EVs that are released by migrating cells. Determining their molecular content and signalling potential clearly need extensive research. The most crucial things to comprehend are how MG works, which signals are triggered by coming into contact with or ingesting MG, what messages they transport between cells, and so on. The discovery of these EVs raises many new questions for future research. In fact, although the heterogeneity of EVs has become obvious, as highlighted by the International Society for Extracellular Vesicles, specific tools to distinguish EVs of different origins are still lacking, and thus different functions are probably not correctly evaluated at the moment.




**O-20. MSC-EVS AS DELIVERY VEHICLES FOR TUMOUR TARGETED THERAPY**

**Róisín Dwyer**
University of Galway, Ireland
**Abstract**
Despite improvements in treatments for breast cancer, when patients are diagnosed with metastatic disease that has spread to distant sites there are limited treatment options and no cure available. Extracellular Vesicles (EVs) hold immense potential as cancer therapeutics due to their small size, biocompatibility, and potential for manipulation of content and surface characteristics. Tumours actively recruit stromal cells including Mesenchymal Stromal Cells (MSCs) into the tumour microenvironment. The tumour-targeted tropism of MSCs is thought to be due to high local concentrations of inflammatory chemokines and growth factors. This tumour tropism combined with the apparent immunosuppressive characteristics of the cells raised remarkable interest in their potential as tumour-targeted delivery vehicles for therapeutic agents. MSC-derived EVs (MSC-EVs) will potentially retain the tumour targeting and immune privilege associated with MSCs while overcoming challenges associated with the use of cells. Our recent work in development of MSC-EVs enriched with a tumour suppressor microRNA for breast cancer therapy will be discussed. The impact of human- and murine-derived MSC-EVs on the immune system and the potential for scale up of EV production will also be highlighted. An approach to support sustained delivery of EVs over time would be very beneficial to prevent cancer resurgence. The use of pre-clinical models that are more reflective of the patient experience will be critical for testing novel approaches to breast cancer treatment. Progress and challenges in the development of MSC-EVs as cancer therapeutics will also be addressed. While we require further understanding of factors mediating EV content, persistence, and uptake, this exciting approach holds tremendous potential for patients with limited existing treatment options.




**O-21. MESENCHYMAL STROMAL CELLS AND THEIR ROLE IN THE TUMOUR MICROENVIRONMENT: CHALLENGES AND OPPORTUNITIES IN THE IMMUNOTHERAPY ERA**

**Andrea Papait**
Cattolica del Sacro Cuore University, Rome, Italy
**Abstract**
Mesenchymal stromal cells (MSCs) have long been studied for their applications in regenerative medicine, exploiting their unique ability to modulate the immune response in a large number of diseases in which the immune response is dysregulated. In addition, MSCs have been studied for their potential applications as an anticancer therapeutic strategy. Indeed, MSCs have been shown to exhibit a natural tropism toward sites of inflammation, and this property has been exploited for their potential use as a vehicle for the selective delivery of anticancer drugs. In addition, MSCs can be genetically modified in an attempt to reactivate the antitumour immune response. Unfortunately, however, the application of MSCs as anticancer therapy has often achieved mixed results. In addition, the fight against cancer must take into account the complex network of interactions that exist between the different actors that are part of the tumour microenvironment (TME) and that are represented by tumour cells, stromal cells, endothelial cells, and immune cells. In fact, the use of MSCs in this regard can be considered a double-edged sword: on the one hand, they can serve as a carrier of drugs or chemotherapeutic compounds; while on the other hand, their immunomodulatory properties can participate in tumour initiation, development, and progression, even contributing to metastasis formation. In this presentation, we will discuss all these merits and demerits of MSCs by contextualising them in the era of immunotherapy, which sees the use of new drugs, mostly monoclonal antibodies, aimed at re-educating the immune response, thus evaluating the possibility of using MSCs or their secretome as adjuvant therapy.




**O-22. INHIBITION OF HUMAN MESOTHELIOMA PROGRESSION IN A MOUSE XENOGRAFT MODEL BY MICRO-FRAGMENTED FAT (MFAT)**

**Valentina Coccè ^1^, Silvia La Monica ^2^, Mara Bonelli ^2^, Giulio Alessandri ^1,3^, Roberta Alfieri ^2^, Costanza Annamaria Lagrasta ^2^, Caterina Frati ^2^, Lisa Flammini ^4^, Aldo Giannì ^1,5^, Luisa Doneda ^1^, Francesco Petrella ^1,6,7^, Francesca Paino ^1^, Augusto Pessina ^1^**




^1^ CRC StaMeTec, Department of Biomedical, Surgical and Dental Sciences, University of Milan, Italy^2^ Department of Medicine and Surgery, University of Parma, Italy^3^ Image Regenerative Clinic, Milan, Italy^4^ Food and Drug Department, University of Parma, Italy^5^ Maxillo-Facial and Dental Unit, Fondazione Ca’ Granda IRCCS Ospedale Maggiore Policlinico, Milan, Italy^6^ Department of Thoracic Surgery, IRCCS European Institute of Oncology, Milan, Italy^7^ Department of Oncology and Hemato-Oncology, University of Milan, Italy




**Objective**
Malignant Pleural Mesothelioma (MPM) is a tumour related to asbestos exposure with no effective therapy and a poor prognosis. Our previous studies demonstrated an in vitro and in vivo inhibitory effect of adipose tissue-derived Mesenchymal Stromal Cells (MSCs) or their derivatives (conditioned media, cellular lysates) on MPM. The purpose of this study was to verify whether fat tissue (FAT), a natural container of MSCs after micro-fragmentation (MFAT) was able to exert a similar inhibitory action on the growth of the human MPM cell line (MSTO-211H) xeno-transplanted in immunodeficient mice.
**Materials and Methods**
MFAT was prepared according to standardised methods using Lipogems device. MSCs were obtained by enzymatic digestion of MFAT. The in vitro effect of MFAT on MSTO-211H cell proliferation was analysed using transwell inserts and measuring the absorbance by a crystal violet assay. PBS were used as negative control. For in vivo experiments, Balb/c-Nude female mice were subcutaneously injected with 10^6^ MSTO-211H cells suspended in Matrigel/PBS. Mice were randomised into 4 groups: control, paclitaxel (PTX), MSCs and MFAT. After a week from injection (time 0), vehicle alone (control group) or PTX (20 mg/kg) were administered intraperitoneally (IP) and MSCs (5 × 10^5^) or MFAT (200 µL) were subcutaneously injected close to the tumour. On days 0, 7, and 14, the size of tumour nodules was measured and on day 20, nodules were collected. Morphometric evaluation of xenograft composition was performed on Masson’s trichrome-stained sections.
**Results**
The in vitro exposure of MSTO-211H cells to MFAT produced a dose-dependent inhibition of cell proliferation. In the in vivo study, the measures of volume of growing tumour mass indicated that the in situ treatment with MFAT produced an important inhibition similar to those obtained in mice treated with the anticancer drug PTX. A trend of inhibition, but not significant, was also observed in mice treated with free MFAT derived MSCs. The morphometric analysis of the tumour xenograft did not show significant differences among groups.
**Conclusions**
Our results show that MFAT, injected in situ, produced a significant (*p* < 0.05) inhibition of the MSTO-211H growth both in vitro and in vivo, and was even comparable to IP PTX treatment. Interestingly, the treatment with free MSCs (5 × 10^5^), at a similar amount contained in around 1 mL of MFAT, exerted only a little anticancer activity.


### 5.2. Posters Communications



**P-01. ANTIMICROBIAL ACTIVITY OF CANINE ADIPOSE TISSUE-DERIVED LYOSECRETOME**

**Valentina Andreoli ^1^, Costanza Spadini ^1^, Mattia Iannarelli ^1^, Priscilla Berni ^1^, Virna Conti ^1^, Roberto Ramoni ^1^, Elia Bari ^2^, Silvia Dotti ^3^, Clotilde Cabassi ^1^, Maria Luisa Torre ^2^ and Stefano Grolli ^1^**




^1^ Department of Veterinary Science, University of Parma, Italy^2^ Department of Pharmaceuticals Sciences, University of Piemonte Orientale, Novara, Italy^3^ Istituto Zooprofilattico Sperimentale della Lombardia e dell’Emilia-Romagna, Brescia, Italy




**Objective**
Mesenchymal Stem/Stromal Cells (MSCs) have been studied and applied as therapeutics in regenerative medicine based on their peculiar properties. Although several papers demonstrate the efficacy of MSC-based therapy in preclinical models, clinical applications are still limited due to doubts about the safety of treatment with viable cells. MSCs secretome is a cell-free product that maintains a large part of cells’ therapeutic properties providing soluble and insoluble bioactive molecules involved in cellular crosstalk. Recent data support an antibacterial activity for both MSCs and their secretome. In this work, canine Lyosecretome (c-Lyo), a freeze-dried secretome prepared from adipose tissue-derived MSCs, has been tested to evaluate its antimicrobial activity against some of the most common canine pathogens involved in infections of the gastrointestinal tract, skin, and ears. Pathogens were subjected to a Minimal Inhibitory Concentration (MIC) assay to evaluate the amount of c-Lyo necessary to inhibit bacterial/fungal growth.
**Materials and Methods**
c-Lyo was resuspended in 500 µL of sterile saline (0.9%) and tested at a concentration range of 20–0.04 mg/mL. Nine replicates for each assay were performed. Tested reference bacteria and yeasts were *E. coli*, *S. Typhimurium*, *S. aureus*, Methicillin-Resistant *S. aureus* (MRSA), *S. pseudintermedius*, *P. aeruginosa,* and *M. pachydermatis*. Strains were amplified for 18–24 h at 37 °C to bring them into logarithmic growth phase and added to the plate at 5 × 10^5^ CFU/mL and incubated at 37 °C overnight. MIC plate reading was performed with a reverse mirror. Mannitol was used as an internal control.
**Results**
A fair inhibitory activity of c-Lyo against both Gram-positive and Gram-negative bacteria was observed. Growth inhibition was higher for Gram-positive (2.2 < MIC < 9.8 mg/mL) but positive results were obtained even on Gram-negative bacteria (10.5 < MIC < 26.1 mg/mL). c-Lyo demonstrated an inhibitory activity also against the yeast *M. pachydermatis* (MIC = 13.3 mg/mL) supporting a possible efficacy in skin infections.
**Conclusions**
MIC data suggest that canine c-Lyo exerts inhibitory activity against various bacterial and yeast pathogens. The availability of an off-the-shelf, ready-to-use MSCs secretome acting as an antibacterial agent could help in replacing/supporting traditional antibiotics therapy, decreasing their use in veterinary medicine, as requested to control the spread of antibiotic resistance.




**P-02. SECRETOME ISOLATED FROM MESENCHYMAL STROMAL CELLS LOADED WITH PACLITAXEL HAVE CYTOTOXIC EFFECT ON OSTEOSARCOMA CELL LINES**

**Alessia Giovanna Santa Banche Niclot ^1^, Elena Marini ^1^, Ivana Ferrero ^2^, Camilla Francesca Proto ^1^, Francesco Barbero ^3^, Ivana Fenoglio ^3^, Alessandro Barge ^4^, Valentina Coccè ^5^, Francesca Paino ^5^, Katia Mareschi ^1,2^ and Franca Fagioli ^1,2^**




^1^ Department of Public Health and Paediatrics, The University of Turin, Italy^2^ Stem Cell Transplantation and Cellular Therapy Laboratory, Paediatric Onco-Haematology Division, Regina Margherita Children’s Hospital, City of Health and Science of Turin, Italy^3^ Department of Chemistry, University of Turin, Italy^4^ Department of Drug Science and Technology, University of Turin, Italy^5^ CRC StaMeTec, Department of Biomedical, Surgical and Dental Sciences, University of Milan, Italy




**Objective**
Bone sarcomas are rare tumours that constitute a very aggressive disease for children and adolescents and represent still an important challenge for clinicians. Our aim is to develop a new method of drug delivery (DDS) based on the Extracellular Vesicles (EVs) loaded with Paclitaxel (PTX) to use for osteosarcoma (OS) treatment. We performed pre-clinical studies to test the cytotoxic effect of Secretome isolated from Mesenchymal Stromal Cells (MSCs) loaded with PTX (PTX-MSCSecr) on OS cell lines (MG63 and SJSA).
**Materials and Methods**
We first calculated the PTX-IC50 in 3 MSC batches, then, we isolated PTX-MSC-Secr by loading MSCs with PTX at the concentration of 15 µg and we analysed its cytotoxic effect on MG63 and SJSA after treatment for 5 days using MTT test. We also analysed the size distribution, particle concentration, and Zeta potential of EVs present in PTX-MSC-Secr by Nanoparticle Tracking Analysis (NTA) instrument. The secretome ability to have EVs with encapsulated PTX was analysed by ultra-high performance liquid chromatography-tandem mass spectrometry (UPLC-MS/MS).
**Results**
PTX-IC50 for MSCs was a mean of 25.1 ± 3.6 µg and for the OS cell lines was, respectively a mean of 17.5 ± 1.0 µg for MG63 and 30 ± 3.1 µg for SJSA. PTX-MSC-Secr induced a decrease of the viability of cells 43 ± 11% and 36 ± 22%, respectively in SJSA and MG63 after five days of treatment. This effect was dose-dependent because scalar dilutions of PTX-MSC-Secr reduced the cytotoxic effect. Cell viability did not decrease after treatment with Secr isolated from CTRL-MSCs (viability of 87 ± 5% in SJSA and of 91 ± 12% in MG63). The dimensional analyses, the result of three independent experiments, indicated EVs mean sizes of 175.8 ± 13 nm (CTRL) and 165.7 ± 11 nm (PTX loaded). EVs Zeta potentials of both CTRL and PTX loaded, as expected, were found to be negative with mean values of −40 ± 2 mV and −38 ± 5 mV.
**Conclusions**
We demonstrated the cytotoxic effect of PTX-MSC-Secr in OS cell lines. The NTA analyses showed a typical mean size of EVs and that no significant differences between the secretome batches and between experimental conditions (CTRL and PTX loaded) were observed. The experiments we performed have provided promising preliminary data that need further investigation but the EVs ability to encapsulate PTX allows us to propose the EVs-PTX isolated from the MSCs as ideal candidates for drug delivery as innovative paediatric sarcomas treatment.




**P-03. 3D BIOPRINTED CONTROLLED RELEASE SCAFFOLD CONTAINING MESENCHYMAL STEM/STROMAL LYOSECRETOME FOR BONE REGENERATION: STERILE MANUFACTURING AND IN VITRO BIOLOGICAL EFFICACY**

**Elia Bari ^1^, Franca Scocozza ^2,3^, Sara Perteghella ^4,5^, Marzio Sorlini ^5,6^, Lorena Segale ^1^, Lorella Giovannelli ^1^, Ferdinando Auricchio ^2,3^, Michele Conti ^2,3^ and Maria Luisa Torre ^4,5^**




^1^ University of Piemonte Orientale, Department of Pharmaceutical Sciences, Novara, Italy^2^ University of Pavia, Department of Civil Engineering and Architecture, Pavia, Italy^3^ P4P S.r.l., Pavia, Italy^4^ University of Pavia, Department of Drug Sciences, Pavia, Italy^5^ PharmaExceed S.r.l., Pavia, Italy^6^ University of Applied Sciences and Arts of Southern Switzerland, SUPSI, Department of Innovative Technologies, Lugano, Switzerland




**Objective**
The present study proposes the design and sterile manufacturing of 3D-printed polycaprolactone (PCL) scaffolds enriched with mesenchymal stem cell (MSC)-secretome for bone tissue regeneration and evaluates their in vitro biological efficacy.
**Materials and Methods**
Adipose MSCs were cultured in DMEM/F12 with 5% platelet lysate (PL); secretome release was obtained by 48 h PL starvation. Supernatants were ultrafiltered, cryoprotectant was added, and freeze-dried, obtaining lyosecretome (0.1 × 10^6^ cell equivalents/mg). Porous parallelepiped-shaped PCL scaffolds (10 × 10 × 3 mm) were prepared by co-printing PCL with an alginate hydrogel (10% *w*/*v*) containing lyosecretome (0.25 mg). Scaffolds were tested for sterility and microbiological tests, as indicated in the EuPh 2.6.27 and 2.6.1 chapters. In addition, the scaffold colonisation by MSCs was investigated by SEM, and in vitro biological efficacy was investigated by MSC osteogenic differentiation and matrix production tests (alizarin red, confocal microscopy, and dosage of osteocalcin by ELISA). Scaffolds without lyosecretome were used as a control.
**Results**
Sterile scaffolds have been obtained and lyosecretome enhanced their colonisation by MSCs: MSCs showed a spread morphology with the initial formation of filopodia, with more frequent and complex cellular processes, overall indicating the cytocompatibility of the scaffold. Lyosecretome also sustained MSC differentiation towards the bone line in an osteogenic medium. Indeed, after 14 days, the amount of mineralised matrix detected by alizarin red was significantly higher for lyosecretome scaffolds. Likely, the amount of osteocalcin, a specific bone matrix protein, was significantly higher at all the times considered (14 and 28 days) for the lyosecretome scaffolds. Confocal microscopy further confirmed such results, demonstrating improved osteogenesis with lyosecretome scaffolds after 14 and 28 days.
**Conclusions**
Overall, these results prove the role of MSC-secretome, co-printed in PCL/alginate scaffolds, in inducing bone regeneration; sterile scaffolds containing MSC-secretome are now available for in vivo preclinical tests of bone regeneration.




**P-04. OSTEOINDUCTIVE AND OSTEOCONDUCTIVE PROPERTIES OF TITANIUM CAGES CONTAINING MESENCHYMAL STEM/STROMAL LYOSECRETOME**

**Elia Bari ^1^, Sara Perteghella ^2,3^, Marzio Sorlini ^3,4^, Delia Mandracchia ^5^, Lorella Giovannelli ^1^, Lorena Segale ^1^ and Maria Luisa Torre ^2,3^**




^1^ University of Piemonte Orientale, Department of Pharmaceutical Sciences, Novara, Italy^2^ University of Pavia, Department of Drug Sciences, Pavia, Italy^3^ PharmaExceed S.r.l., Pavia, Italy^4^ University of Applied Sciences and Arts of Southern Switzerland, SUPSI, Department of Innovative Technologies, Lugano, Switzerland^5^ University of Brescia, Department of Molecular and Translational Medicine, Brescia, Italy




**Objective**
The present study investigates the capacity of the mesenchymal stem cell (MSC)-secretome, formulated as a ready-to-use and freeze-dried medicinal product (the lyosecretome), to promote the osteoinductive and osteoconductive properties of titanium cages.
**Materials and Methods**
Adipose MSCs were cultured in DMEM/F12 with 5% platelet lysate (PL); secretome release was obtained by 48 h PL starvation. Supernatants were ultrafiltered, cryoprotectant was added, and freeze-dried, obtaining lyosecretome (0.1 × 10^6^ cell equivalents/mg). Lyosecretome was added to titanium cages (1 × 1 × 0.3 cm in size) kindly provided by MT Ortho and manufactured by an additive manufacturing technology called electron beam melting. The cages colonisation by MSCs was investigated by SEM, and in vitro biological efficacy was investigated by MSC osteogenic differentiation and matrix production tests (alizarin red, confocal microscopy and dosage of osteocalcin by ELISA). Cages without lyosecretome were used as a control.
**Results**
After 14 days, in the presence of lyosecretome, significant cell proliferation improvement was observed. Scanning electron microscopy revealed the cytocompatibility of titanium cages: the MSCs seeded showed a spread morphology and the initial formation of filopodia. After seven days, in the presence of lyosecretome, more frequent and complex cellular processes forming bridges across the porous surface of the scaffold were revealed. Moreover, after 14 and 28 days of a culture in an osteogenic medium, the amount of mineralised matrix detected by alizarin red was significantly higher when lyosecretome was used. Finally, improved osteogenesis with lyosecretome was confirmed by confocal analysis after 28 and 56 days of treatment and demonstrating the production by osteoblast-differentiated MSCs of osteocalcin, a specific bone matrix protein.
**Conclusions**
Overall, these results confirm the role of MSC-secretome in combination with titanium cages in inducing bone regeneration. Such scaffold prototypes for bone regenerative medicine are now available for further in vivo safety and efficacy testing.




**P-05. OSTEOINDUCTIVE AND OSTEOCONDUCTIVE PROPERTIES OF BIOHYBRID BOVINE MATRIX CONTAINING MESENCHYMAL STEM/STROMAL LYOSECRETOME**

**Elia Bari ^1^, Ilaria Roato ^2^, Giuseppe Perale ^3,4,5^, Filippo Rossi ^6^, Tullio Genova ^7^, Federico Mussano ^2^, Riccardo Ferracini ^8^, Marzio Sorlini ^9,10^, Sara Perteghella ^1,10^ and Maria Luisa Torre ^1,10^**




^1^ University of Piemonte Orientale, Department of Pharmaceutical Sciences, Novara, Italy^2^ University of Torino, Department of Surgical Sciences, CIR-Dental School, Torino, Italy^3^ Industrie Biomediche Insubri SA, Mezzovico-Vira, Switzerland^4^ University of Southern Switzerland (USI), Faculty of Biomedical Sciences, Lugano, Italy^5^ Ludwig Boltzmann Institute for Experimental and Clinical Traumatology, Vienna, Austria^6^ Politecnico di Milano, Department of Chemistry, Materials and Chemical Engineering Milan, Italy^7^ University of Torino, Department of Life Sciences and Systems Biology, Turin, Italy^8^ University of Genova, Department of Surgical Sciences and Integrated Diagnostics, Genoa, Italy^9^ University of Applied Sciences and Arts of Southern Switzerland, SUPSI, Department of Innovative Technologies, Lugano, Switzerland^10^ PharmaExceed Srl, Pavia, Italy




**Objective**
The present study combines biohybrid bone substitute scaffolds (SB) with lyosecretome, a freeze-dried MSC-secretome formulation containing proteins and extracellular vesicles and evaluates the osteoinductive and osteoconductive in vitro.
**Materials and Methods**
Adipose MSCs were cultured in DMEM/F12 with 5% platelet lysate (PL); secretome release was obtained by 48 h PL starvation. Supernatants were ultrafiltered, cryoprotectant was added, and freeze-dried, obtaining Lyosecretome (0.1 × 10^6^ cell equivalents/mg). Each SB scaffold (1 × 1 × 0.3 cm in size) was loaded with 16 × 10^3^ cell equivalents of Lyosecretome by an absorption method, obtaining SBlyo. 1 × 10^6^ MSCs were seeded onto the upper surface of SB in an osteogenic medium, and after 14 and 60 days of cultures, gene expression for osteocalcin (OCN), alkaline phosphatase (ALP), and collagen 1 (COLL-1) was evaluated. After 60 days, a high-resolution X-ray microtomography was used to identify the newly formed mineralised tissue after cell colonisation on SB and SBlyo. Moreover, SB and SBlyo were fixed, decalcified, dehydrated, cut into thin sections, and stained with hematoxylin and eosin (H&E) for morphological analyses. Immunohistochemical analysis was also performed using antibodies for osteocalcin and TGF-β.
**Results**
After 14 days, significant cell proliferation improvement was observed on SBlyo with respect to SB, where cells filled the cavities between the native trabeculae. For SB, on the other hand, the process was still present, but tissue formation was less organised at 60 days. On both scaffolds, cells differentiated into osteoblasts and were able to mineralise after 60 days. SBlyo showed a higher expression of osteoblast markers and a higher quantity of newly formed trabeculae than SB alone. The quantification analysis of the newly formed mineralised tissue and the immunohistochemical studies demonstrated that SBlyo induces bone formation more effectively. This osteoinductive effect is likely due to the osteogenic factors in the lyosecretome, such as fibronectin, alpha-2-macroglobulin, apolipoprotein A, and TGF-β.
**Conclusions**
Overall, these results confirm the role of MSC-secretome loaded on biohybrid bovine matrix in inducing bone regeneration. Such scaffold prototypes for bone regenerative medicine are now available for further in vivo safety and efficacy testing.




**P-06. LOCAL AND SYSTEMIC APPLICATION OF AUTOLOGOUS MESENCHYMAL STROMAL CELLS IN CATS SUFFERING FROM CHRONIC GINGIVOSTOMATITIS: A PILOT STUDY**

**Priscilla Berni ^1^, Tommaso Magni ^2^, Maurizio Del Bue ^3^, Virna Conti ^1^, Valentina Andreoli ^1^, Rosanna Di Lecce ^1^, Anna Maria Cantoni ^1^, Roberto Ramoni ^1^, Stefano Grolli ^1^**




^1^ Department of Veterinary Medical Science, University of Parma, Italy^2^ Veterinary Practitioner, Clinica Veterinaria Pet Care, Bologna, Italy^3^ Freelance Veterinary Medical Doctor, Parma, Italy




**Objective**
Feline Chronic Gingivostomatitis (FCGS) is a severe inflammatory oral disease characterised by painful mucosal lesions, oral discomfort, inappetence, reduced grooming, weight loss, and hypersalivation, seriously affecting the patient’s quality of life. The current standard of care is invasive full/near-full mouth tooth extraction and long-term pharmacological treatments, with a high rate of relapse. Since FCGS is probably immune-mediated, Mesenchymal Stromal Cells (MSCs) represent a promising tool for this disorder. Different studies have reported the efficacy of systemic administration of adipose-derived MSCs (Ad-MSCs) in cats with FCGS, while a pilot study reported a lack of efficacy when the treatment is performed prior to full-mouth tooth extraction. This study aims to determine the efficacy of local and systemic administration of Ad-MSCs in cats with FCGS, with or without teeth.
**Materials and Methods**
Eleven client-owned cats with FCGS and with long-term pharmacological clinical history, with or without teeth, were treated with a double application of autologous Ad-MSCs at 30-day intervals. The cats were enrolled in two groups: one was treated with local injections of 5 × 10^6^ autologous Ad-MSCs and the other was treated with local injections associated with systemic infusions of 2 × 10^6^/Kg autologous Ad-MSCs. An oral examination with photographs and oral biopsies was performed at the enrolment and 30 days after each treatment. A SDAI (Stomatitis index) scoring was calculated at the same intervals, in addition to a brief owner questionnaire and a veterinarian scoring. Furthermore, a complete blood count, blood immune cell phenotyping, and biochemical profile were planned on day zero and three months after the first treatment.
**Results**
At the time of writing, eight cats have been treated with double MSCs application. Seven cats have completely suspended any pharmacological treatment after the first application. The clinical assessment at day 60 showed a marked clinical improvement reported by the owners, except for one patient that showed the maximum SDAI score at the enrolling who improved only in the body weight parameter. A statistically significant difference was observed in the SDAI between day 0 and 60 for seven cats, two with a complete resolution of the oral inflammation (*p* < 0.05).
**Conclusions**
Immunohistochemical analysis and blood immune cell phenotyping are needed to confirm the observed clinical improvement.




**P-07. A NEW PROTOCOL FOR VALIDATION OF CHONDRO, ADIPO AND OSTEO DIFFERENTIATION KIT OF CULTURED ADIPOSE-DERIVED STEM CELLS (ADSC) BY REAL-TIME RT-QPCR**

**Carlotta Castagnoli ^1^, Valentina Daprà ^2,3^, Daniela Alotto ^1^, Stefania Casarin ^1^, Stefano Gambarino ^2,3^, Mara Fumagalli ^1^, Sara Castiglia ^4^, Deborah Rustichelli ^4^, Maddalena Dini ^3^, Ilaria Galliano ^2^, Massimiliano Bergallo ^2,3^**




^1^ Skin Bank, Department of General and Specialized Surgery, University Hospital City of Health and Science of Turin, Italy^2^ Department of Public Health and Pediatric Sciences, University of Turin, Italy^3^ BioMole srl, Spin-off University of Turin, Italy^4^ Stem Cell Transplantation and Cellular Therapy Laboratory, Paediatric Onco-Haematology Division, Regina Margherita Children’s Hospital, City of Health and Science of Turin, Italy




**Objective**
Mesenchymal stem cells (MSCs) are multipotent cells, originally derived from the embryonic mesenchyme, and able to differentiate into connective tissues such as bone, fat, cartilage, tendon, and muscle. Furthermore, MSCs derived from adipose tissue ADSC (Adipose-derived Stem Cells) show great potential for degenerative disease treatment.In this study, we designed a series of experiments based on real-time rt-QPCR to validate a new commercially available kit able to explore changes in gene expression in human ADSC subjected to osteogenic, adipogenic, and chondrogenic differentiation.
**Materials and Methods**
As the primary outcome of the study, we selected better indicators of trilineage differentiation using the third passage of cultured ADSC isolated from the stromal vascular fraction (SVF) by enzymatic digestion. ACAN, FABP4A, and Col11a1 were selected as indicators of chondrogenic, adipogenic, and osteogenic differentiation, respectively based on statistically significant results.As a secondary outcome of the study, an in vitro ageing test was performed from passage 2 to 6 to evaluate the highest differentiation potential. Total RNA extraction from differentiated and control ADSC were performed. Relative quantifications of mRNA expression of selected genes was completed according to rt-PCR kit protocol.
**Results**
ACAN detection, a test for chondrogenic differentiation, revealed equivalent ∆∆Ct values between P3 and P6. These were considered equivalent passages for induction differentiation tests. FABP4 detection, an assay for adipogenic differentiation, showed similar results for all cell passages tested so they can all be considered suitable in differentiation assay induction; on the contrary, only passage P6 for Col11a1 was suitable for osteogenic differentiation.
**Conclusions**
In conclusion, we validated a new real-time rt-QPCR protocol able to evaluate osteogenic, chondrogenic, and adipogenic ADSC differentiation in vitro.




**P-08. EVALUATION OF THE EFFECTS OF HUMAN AMNIOTIC MESENCHYMAL STROMAL CELLS AND AMNIOTIC MEMBRANE CONDITIONED MEDIUM ON OVARIAN CANCER CELLS USING 2D AND 3D CELL MODELS**

**Paola Chiodelli ^1^, Patrizia Bonassi Signoroni ^1^, Silvia De Munari ^1^, Andrea Papait ^2^, Sara Ficai ^2^, Antonietta Silini ^1^ and Ornella Parolini ^2,3^**




^1^ Centro di Ricerca E. Menni, Fondazione Poliambulanza Istituto Ospedaliero, Brescia, Italy^2^ Department of Life science and Public Health, Università Cattolica del Sacro Cuore, Rome, Italy^3^ Fondazione Policlinico Universitario “Agostino Gemelli” IRCCS, Rome, Italy




**Objective**
Ovarian cancer is the seventh most common cancer and sixth most common cause of cancer death for women globally. Nowadays, surgical resection followed by chemotherapy is the standard of care. However, a number of patients are faced with recurrence due to tumour dissemination and acquired chemoresistance. Therefore, the novel alternative approaches targeting ovarian cancer cells are urgently needed to improve the standard of care for patients. With this regard, mesenchymal stromal cells (MSC) constitute a compelling therapeutic option. Of particular interest, MSC isolated from the amniotic membrane of the human term placenta (hAMSC) are of therapeutic interest and present noteworthy advantages when compared to MSC from other sources, such as their ease of recovery from biological waste. In addition, we previously reported that the hAMSC secretome has antiproliferative effects in vitro when co-cultured with different tumour cells.
**Material and Methods**
We decided to evaluate the possible anti-proliferative effects of the secretome (conditioned medium, CM) from hAMSC in 2D and 3D models of ovarian cancer. In parallel, we evaluated the CM from the intact amniotic membrane (hAM) to see if antiproliferative effects could be maintained without the need to perform MSC isolation.
**Results**
Both CM-hAMSC and CM-hAM inhibit the proliferation and migration of ovarian cancer cells (HEY and SKOV-3) in 2D scratch assays. Moreover, both CM-hAMSC and CM-hAM affect the apoptotic process in HEY and SKOV-3. In the SKOV-3 spheroid 3D model, CM-hAMSC and CM-hAM significantly decrease the spheroid area inducing also a change in spheroid morphology.
**Conclusions**
The data so far collected indicated that CM-hAMSC and CM-hAM impact the growth and activity of ovarian cancer cells. Further experiments are needed to better understand their inhibitory capacity on ovarian cancer cells in 2D and in 3D models and clarify their use as a potential adjuvant therapeutic strategy able to target tumour cells.




**P-09. EXTRACELLULAR VESICLES FROM CORD BLOOD MESENCHYMAL STROMAL CELLS STIMULATE REGENERATION PROCESS IN ORGANOTYPIC MODEL OF NEURONAL INJURY**

**Stefania D’Agostino ^1,2^, Francesca Cecchinato ^2,4^, Beatrice Auletta ^2,4^, Marcin Jurga ^3^, Maurizio Muraca ^1,2^, Anna Urciuolo ^2,4^, Michela Pozzobon ^1,2^**




^1^ Department of Women’s and Children’s Health, University of Padova, Italy^2^ Institute of Pediatric Research (IRP) Città della Speranza, Padova, Italy^3^ EXO Biologics (SA), Belgium^4^ Department of Molecular Medicine, University of Padova, Italy




**Objective**
The central nervous system (CNS) has only a limited capacity to regenerate, hence, after injury, a progressive loss of neurons, due to homeostatic imbalance, leads to neurodegenerative pathology. The homeostasis process critically depends on the interaction between neurons and glial cells.Novel treatment suggestions for neurodegenerative disorders consider the use of cell-derived products, relying on the beneficial paracrine effects of the applied products.Extracellular vesicles (EVs) recently emerged as versatile messengers in CNS cell communication. These nanoparticles, defined by a phospholipid bilayer, can convey signals by triggering surface receptors, activating second messenger signalling cascades or delivering their cargo, such as proteins, nucleic acids, and small molecules.In this work, we considered an organotypic in vitro model of spinal injury treated with EVs derived by Wharton Jelly Mesenchymal stromal cells.
**Materials and Methods**
Isolated spinal cords from rat foetuses were cut into small pieces and cultured on a bed of Matrigel. Organotypic spinal cord (oSpC) were treated with EVs soon after the beginning and after 24 h of culture. To monitor the early effect of the EVs on neural axon sprouting, samples were analysed 48 h after the seeding.
**Results**
Our results showed that neural axon sprouting was significantly increased in EV-treated samples when compared to untreated oSpCs. Moreover, the neural protein TujI/TUBB3 expressed in the developing neurons and the glia marker GFAP, that identified new astrocytes, were differently detected in the presence of EVs. Fluo-4 imaging revealed a more controlled calcium flux in oSpC treated with EVs at the axonal projection compared to the untreated SpC.Real-time PCR on neural miRNA highlighted how the EVs can be responsible for miRNA mediator of neural regeneration.
**Conclusions**
Our results suggest a possible role of perinatal EVs in promoting neural axon sprouting, opening new perspectives for their application in neural regeneration, and as new exciting signalling modalities that add a new dimension to the interaction between neurons and glial cells.




**P-10. 3D ORGANOIDS: A NEW TRANSLATIONAL APPROACH FOR THE ASSESSMENT OF THE IMMUNOMODULATORY ACTIVITY OF MESENCHYMAL STROMAL CELLS DERIVED EXTRACELLULAR VESICLES IN INFLAMMATORY BOWEL DISEASE**

**Giada de lazzari ^1,2,3,5,6^, David Sagnat ^5,6^, Ricardo Malvicini ^1,2,3^, Michela Pozzobon ^1,2^, Gustavo Yannarelli ^4^, Anna Maria Tolomeo ^1,2,3^, Nathalie Vergnolle ^5,6,7^, Maurizio Muraca ^1,2,3^**




^1^ Foundation Institute of Pediatric Research Città della Speranza, Padova, Italy^2^ Dept of women’s and children’s health, University of Padova, Italy^3^ L.I.F.E.L.A.B. Program, Consorzio per la Ricerca Sanitaria (coris), Veneto Region, Padova, Italy^4^ Instituto de Medicina Traslacional, Trasplante y Bioingeniería (imettyb-conicet), Buenos Aires, Argentina^5^ Toulouse Organoid Platform, France^6^ Inserm, Toulouse, France^7^ University of Calgary, Faculty of Medicine, Department of Physiology and Pharmacology, Calgary, AB, Canada




**Objective**
Inflammatory bowel diseases (IBD), including Crohn’s disease (CD) and Ulcerative Colitis (UC), are chronic relapsing–remitting disorders characterised by inflammation of the gut. Different factors contribute to IBD development, such as deficiencies in epithelial integrity, immune response mechanisms, and mucosal barrier function, whose complexity is difficult to reproduce in experimental conditions.The need to overcome the typical limits of cell lines, animal models, and organ culture, led to the development of a 3D culture system capable of maintaining the characteristics of the target organ in the long term. We thus explored the feasibility of this tool to evaluate the therapeutic potential of extracellular vesicles derived from mesenchymal stromal cells (MSC-EVs) whose immunomodulatory activity is the object of increasing interest.
**Materials and Methods**
MSC-EVs were isolated from human umbilical cord MSCs by ultrafiltration (100 kD cutoff), quantified by TRPS and characterised for established markers by MACSPlex.Colon organoids were derived from human colon samples’ extracted crypts and seeded in Matrigel beads. Inflamed (IBD) organoids, in particular, were obtained from the stimulation of control organoids with a pro-inflammatory cocktail (IL-1β, TNF-α, and IL-6, at 100 ng/mL) on days 3, 5, and 7 to reproduce a chronic inflammatory state.To evaluate the effect of MSC-EVs, IBD organoids were then treated on day 7 with a dose of 1 × 10^9^ MSC-EVs/mL at different time points (3 h, 6 h, 24 h, and 48 h).As readouts, we evaluated the effect on proliferation, differentiation, inflammation, barrier, and growth by immunofluorescence and molecular biology.
**Results**
We found that the stimulation with the pro-inflammatory cocktail increases inflammation (IL8, MCP1, and TNF- α), stress (Chop), carcinogenesis (Wnt5a), growth (OCT4, SOX9), proliferation (EGF), and decreases differentiation (MUC2) and epithelial barrier marker (Epcam) compared to control organoids. Treatment with MSC-EVs at 24 h decreased inflammation, stress, growth, proliferation, carcinogenesis, and increased differentiation and epithelial barrier marker.
**Conclusions**
In conclusion, we have provided preliminary evidence of the therapeutic effect of MSC-EVs in inflamed intestinal organoids. These results suggest that the present 3D model could represent a useful experimental tool closely reproducing some critical features of human IBD.




**P-11. CHARACTERIZATION OF HUMAN DENTAL PULP STEM CELLS MULTICELLULAR SPHEROIDS AS ORGANOTYPIC THREE-DIMENSIONAL IN VITRO MODEL**

**Ilaria De Santis ^1,2^, Alessandro Bevilacqua ^3,4^, Thimios A. Mitsiadis ^2^, Deborah Stanco ^2^**




^1^ University of Bologna, Interdepartmental Centre Alma Mater Research Institute on Global Challenges and Climate Change (Alma Climate), Bologna, Italy^2^ University of Zürich, Orofacial Development and Regeneration, Institute of Oral Biology, Zürich, Switzerland^3^ University of Bologna, Advanced Research Center on Electronic Systems (ARCES) for Information and Communication Technologies “E. De Castro”, Bologna, Italy^4^ University of Bologna, Department of Computer Science and Engineering (DISI), Bologna, Italy




**Objective**
Dental pulp stem cells (DPSCs) are mesenchymal stem cells (MSCs) of neural crest origin. High availability, multipotency, and plasticity make them a promising source of patient-specific MSCs for personalised regenerative medicine strategies. However, their clinical translation still faces many challenges due to a lack of deep understanding of their niche microenvironment, biology, and functionality in vivo. By recapitulating the complex in vivo-like microenvironment, three-dimensional (3D) multicellular spheroids open to new opportunities for MSCs translation in clinical and preclinical settings. In this context, the development of human DPSC multicellular spheroids as organotypic 3D in vitro models was assessed.
**Material and Methods**
2 × 10^4^ DPSCs at passage IV were used for spheroid creation by hanging drop technique and transferred to 96 ULA plates after 24 h. Spheroid morphology, viability (FDA-PI staining), and metabolic activity (Alamar Blue assay) were evaluated at day 1, 2, 3, 4, and 7 of culture. Tissue-specific (nestin, vimentin, collagen I and IV, laminin, and fibronectin) and stem-related (BMI1, CD90, SOX2, NANOG, and OCT4) markers were evaluated at gene (qRT-PCR) and protein level (IF). After bright field imaging and by on-purpose method for automated image analysis, spheroid dimension, morphology, and compactness were quantified by their equivalent diameter (ED), sphericity (S), and border indentation (BI), respectively. After normality check by Shapiro–Wilk test, statistical significance was assessed by *t*-test or Wilcoxon test (*p* < 0.05).
**Results**
The hanging drop technique has a mean efficiency of spherical (S ≥ 80%) spheroid creation at 82% in 24 h. ED progressively shrinks, while S rises up to day four. At day three, spheroids show highest viability (I_FDA_/I_PI_ = 12.8, *p* < 10^−6^) and the most stable morphology (ED = 488 ± 43 µm, S = 0.87 ± 0.04, BI = 0.79 ± 0.03). Spheroid metabolic activity was stable in time and significantly lower than standard 2D culture (−35% avg, *p* < 0.03). DPSC spheroids expressed high levels of nestin, vimentin, collagen I and IV, laminin, and fibronectin, beside significantly higher mRNA levels of BMI1, CD90, SOX2, NANOG, and OCT4.
**Conclusions**
The human DPSC spheroids can be easily and quickly created by a low-cost procedure in 24 h and they may efficiently enhance the therapeutic action of DPSCs. Moreover, automated image analysis here proves as a valuable tool for the finest analysis of DPSC spheroid morphology in future preclinical setting applications.




**P-12. CELL-FREE ORTHOBIOLOGICS FROM ADIPOSE MESENCHYMAL STEM/STROMAL CELLS: THE ROAD SO FAR AND FUTURE PERSPECTIVES IN OSTEOARTHRITIS TREATMENT**

**Elena Della Morte ^1^, Chiara Giannasi ^1,2^, Stefania Niada ^1^ and Anna Teresa Brini ^1,2^**




^1^ IRCCS Istituto Ortopedico Galeazzi, Milan, Italy^2^ University of Milan, Department of Biomedical Surgical and Dental Sciences, Milan, Italy




**Objective**
Mesenchymal Stem/stromal Cells, and, in particular, adipose-derived ones (ASCs), show great therapeutic potential in counteracting orthopaedic conditions. Since a large part of ASC action is exerted through paracrine signalling, in the last years, we focused on the study of their conditioned medium (CM) in terms of molecular composition and biological action on experimental models of osteoarthritis (OA).
**Materials and Methods**
The CM was obtained from confluent ASCs cultured for 72 h in the absence of FBS, then concentrated through filter devices with a 3 kDa cut off. Its components were investigated through Raman Spectroscopy, NTA, -omic approaches, and ELISA. Articular chondrocytes (CHs) and osteochondral explants (OEs) were harvested from patients undergoing arthroplasty at IRCCS Istituto Ortopedico Galeazzi prior to approval of the ethics committee. OA phenotype was induced by stimulation with 10 ng/mL TNFα, and specimens were treated with the CM derived from 0.5–1 × 10^6^ ASCs. The levels of inflammatory, hypertrophic, and catabolic factors were monitored through time by immunological or enzymatic assays.
**Results**
ASC-CM peculiar Raman fingerprint and its characteristic vesicular profile were depicted. The analysis of ASC-CM composition showed the presence of stable levels of bioactive factors that can be putative players in counteracting the OA process. Among other potential effectors, the abundance of the chondroprotective factors Dkk-1 and TIMP-1/2 was particularly intriguing. Accordingly, in both experimental OA models, i.e., articular chondrocytes (CHs) and osteochondral explants (OEs), ASC-CM efficiently buffered the TNFα-induced aberrant activity of matrix metalloproteinases. Furthermore, ASC-CM reverted TNFα-induced expression of PGE2 and COL10A1 in CHs, while it lowered the catabolic release of glycosaminoglycans on OEs.
**Conclusions**
These in vitro and ex vivo experiments confirm ASC-CM beneficial potential in dampening OA-related hallmarks, encouraging deeper investigations of this product in the perspective of its future clinical translation as a cell-free orthobiologic.




**P-13. LATTICE-BASED SCAFFOLDS FOR THE BULK REINFORCEMENT OF SOFT MATERIALS FOR CARTILAGE REGENERATION**

**Stephanie E. Doyle ^1,2^, Finn Snow ^1^, Rhyys Turner ^1^, Carmine Onofrillo ^2,3,4^, Cathal D. O’Connell ^1,2^, Claudia Di Bella ^2,3,5^, Elena Pirogova ^1^, Serena Duchi ^2,3,4^**




^1^ Electrical and Biomedical Engineering, School of Engineering, RMIT University, Melbourne, Victoria, Australia^2^ ACMD, St Vincent’s Hospital Melbourne, Fitzroy, Victoria, Australia^3^ Department of Surgery, The University of Melbourne, St Vincent’s Hospital Melbourne, Fitzroy, Victoria, Australia^4^ ARC Centre of Excellence for Electromaterials Science, Intelligent Polymer Research Institute, University of Wollongong, Wollongong, NSW, Australia^5^ Department of Orthopaedics, St Vincent’s Hospital Melbourne, Fitzroy, Victoria, Australia




**Objective**
Hydrogels are a fundamental element of cartilage engineering by providing a suitable environment for cells to proliferate, migrate, and differentiate. However, this typically soft environment is often not suitable under high mechanical loads. This can present an issue for implantable scaffolds where a large stiffness mismatch between the implant and native tissue can contribute to its failure. With this work we prioritise the bulk mechanical properties of the implant to reduce the stiffness mismatch with the native tissue. We aim to do this without negatively impacting the environment, previously established by our team, for mesenchymal stem/stromal cells (MSCs) to create new cartilaginous tissue.
**Materials and Methods**
Lattice scaffolds, designed to minimise the total volume fraction of the reinforcement scaffold, are made via the Negative Embodied Sacrificial Template (NEST) 3D printing process from polycaprolactone. The NEST scaffolds are combined with MSCs of adipose origin and embedded in a soft, 6% gelatin methacryloyl (GelMA) hydrogel and UV photo crosslinked. After six weeks of culture, samples are analysed by a combination of imaging (immunohistochemistry and histology staining), metabolic activity, glycosaminoglycan (GAG) content, compression testing, and quantitative polymerase chain reaction (qPCR).
**Results**
Using highly porous (up to 90%) lattice designs we have demonstrated the biocompatibility of the NEST scaffolds where we recorded no significant difference in metabolic activity between reinforced and control samples after six weeks in in vitro culture. Secondly, limited to no interference of the NEST scaffold on bulk cell differentiation as measured by GAG production, qPCR, and imaging. Third, the ability to match the native tissue stiffness from day 0, for example, native articular cartilage of the knee has a compressive modulus of ≈400–800 kPa and our reinforced NEST scaffold of a clinically relevant size embedded in 6% GelMA has a compressive modulus of ≈480 kPa.
**Conclusions**
Our reinforced hydrogels retain the ideal soft environment for cells to differentiate down the chondrogenic lineage while reducing the stiffness mismatch between the implant and native tissue. Therefore, we address one area, stiffness mismatch, where implants can fail when moving from an in vitro model to in vivo.




**P-14. MESENCHYMAL STEM CELLS AS A SYSTEM FOR DELIVERING NAB-PACLITAXEL IN A METASTATIC MODEL OF PDAC**

**Benedetta Ferrara, Smeralda Rapisarda, Antonio Citro, Martina Policardi, Fabio Manenti, Chiara Gnasso, Annapaola Andolfo, Denise Drago, Lorenzo Piemonti**
Diabetes Research Institute, IRCCS San Raffaele, Milan, Italy
**Objective**
Current therapy for metastatic pancreatic ductal adenocarcinoma (PDAC) is limited by drug toxicity and resistance. A delivery system might improve drug bioavailability. The aim of this study was to investigate mesenchymal stem cells (MSCs) for delivering nab-paclitaxel (nPTX) in metastatic PDAC. The feasible manipulation of MSCs, their ability to deliver drugs and to do homing in the damaged site make them a suitable candidate for an autologous cell-therapy approach.
**Materials and Methods**
An in vivo model of liver metastases was generated by injection of K8484 murine PDAC cells derived from KPC mice into the portal vein of C57BL/6N mice. Firstly, tumour-bearing mice were treated with nPTX to assess their sensitivity to this drug by monitoring the metastatic volume by magnetic resonance. Secondly, the MSCs viability after treatment with nPTX, as well as drug uptake and release, were investigated in vitro. The biodistribution of luciferase (LUC)-transduced MSCs intraportally injected in tumour-bearing mice was investigated by in vivo imaging systems (IVIS). Finally, the effect on the metastatic reduction of nPTX-loaded MSCs was evaluated on tumour-bearing mice after intraportal injection.
**Results**
NPTX significantly reduced the metastatic volume of tumour-bearing mice, demonstrating the responsiveness of our model to this drug. In vitro, after 24 h, we could appreciate a high uptake of nPTX by MSCs without a reduction of cell viability. NPTX release by MSCs was higher after 24 h but sustained until 72 h, as reported by mass-spectrometry analysis. Biodistribution studies revealed a high and prolonged accumulation of MSCs in the liver after intraportal injection. Results obtained in vivo on the antitumour efficacy of this system reported a significantly higher metastatic reduction in mice treated with nPTXloaded MSCs with respect to control mice treated with not loaded MSCs or free nPTX.
**Conclusions**
The ability of MSCs to incorporate nPTX to a high extent and without reporting toxicity is promising. This system allowed to lower the drug dose, thus reducing nPTX toxicity and specifically targeting the tumour site reporting an effective reduction of the metastatic burden. Obtained results open new insights into the use of MSCs for delivering nPTX as a suitable and promising therapeutic option for PDAC.




**P-15. DISSECTING THE EFFECT OF MICROPLASTICS ON HUMAN AMNIOTIC MESENCHYMAL STROMAL CELLS**

**Sara Ficai ^1^, Andrea Papait ^1,2^, Alice Masserdotti ^1^, Patrizia Bonassi ^3^, Antonietta Silini ^3^, Ornella Parolini ^1,2^**




^1^ Department of Life Science and Public Health, Università Cattolica del Sacro Cuore, Rome, Italy^2^ Fondazione Policlinico Universitario “Agostino Gemelli” IRCCS, Rome, Italy^3^ Centro di Ricerca E. Menni, Fondazione Poliambulanza Istituto Ospedaliero, Brescia, Italy




**Objective**
Environmental microplastic (MPs, 1 µm–5 mm) degradation has become a problem for human health due to the possible production of toxic metabolites, contaminating water, air, food, and several daily used products. Bisphenol A (BPA) is the most representative chemical component of MPs debris, and it is reported to alter cellular functions by acting as an endocrine disruptor. Endocrine disrupting chemicals (EDC) are able to interfere with endogenous hormone biosynthesis, metabolism, and functions through the binding with typical and peculiar oestrogen receptors, triggering a dysregulation in cellular physiological processes such as oxidative stress. Placental tissues are supposed to be susceptible to EDC for the abundance of hormone receptor expression. Thus, it has been hypothesised that exposure of the mother to MPs-derived chemicals, such as BPA, can lead to an imbalance of the physiological processes that contribute to a successful pregnancy, increasing the risk of gestation-related complications. Based on this evidence, our purpose is to evaluate the effects of BPA on mesenchymal stromal cells isolated from the amniotic membrane of the human term placenta (hAMSC).
**Materials and Methods**
Our preliminary experiments aimed to evaluate the impact of BPA on hAMSC properties and functions in vitro. Cellular viability, metabolism, and apoptotic rate were analysed after a 24 h exposure to increasing concentrations of BPA (0.1; 0.2; 0.3; 0.4 µM) by MTT assay, ATP lite assay, and PI/Annexin kit, respectively. Concomitantly, the loss of cellular oxidative balance has been assessed by flow cytometry 3 h and 24 h after BPA exposure. In addition, dysregulation in the expression of cell cycle mediators was evaluated by RT-PCR.
**Results**
We first observed a dose-dependent reduction in hAMSC viability and metabolic capacity as well as an enhancement in cellular apoptotic rate after the treatment with increasing concentrations of BPA. At the highest concentration of BPA used in our study, hAMSC-intracellular oxidative stress and the gene expression of typical cell cycle regulators both increased.
**Conclusions**
Our preliminary data suggest that BPA may affect hAMSC functions. In light of these observations, additional studies will be performed to evaluate intracellular pathways through which BPA can act, supposing that adverse consequences on placenta resident MSC may be related to a negative outcome of gestation and riskiness for the baby.




**P-16. HINTS ON THE METABOLIC STATUS OF SERUM-STARVED MESENCHYMAL STEM/STROMAL CELLS**

**Chiara Giannasi ^1,2^, Stefania Niada ^2^, Elena Della Morte ^2^ and Anna Teresa Brini ^1,2^**




^1^ University of Milan, Department of Biomedical Surgical and Dental Sciences, Milan, Italy^2^ IRCCS Istituto Ortopedico Galeazzi, Milan, Italy




**Objective**
In the last decade, the scientific interest in the secretome of Mesenchymal Stem/stromal Cells (MSCs) has increased tremendously due to its promising potential as an alternative to cell therapy. Mid-term serum starvation represents a convenient strategy for secretome production since it stimulates cell secretion while reducing the drawbacks associated with the use of animal-derived supplements. Nevertheless, the impact of this procedure on the metabolic status of donor cells still needs to be defined. Here, we investigate this aspect through metabolomics by comparing MSCs cultured with 10% foetal bovine serum (FBS) and serum-starved ones.
**Materials and Methods**
Primary human adipose-derived MSCs were grown in complete culture medium until confluence. Then, cells were either collected and analysed or rinsed and cultured for three days in the absence of FBS. Samples were screened for polar and apolar molecules by untargeted metabolomics at the Proteomics and Metabolomics Facility of IRCCS Ospedale San Raffaele. The differences revealed by this analysis were further validated by ad hoc biochemical and functional assays.
**Results**
Differential metabolomics shows a clear clustering between samples grown in standard conditions and under serum deprivation. Metabolite set enrichment analysis reveals several processes affected by serum withdrawal, most of them occurring at the mitochondrial level such as Mitochondrial Electron Transport Chain, Oxidation of Branched Chain Fatty Acids, and Citric Acid Cycle. The impairment of mitochondrial metabolism is further confirmed by the significant accumulation of reactive oxygen species and the reduction of succinate dehydrogenase activity. At last, cells exposed to serum starvation show higher expression levels of mitochondrial superoxide dismutase.
**Conclusions**
Mid-term serum deprivation affects cell metabolomes by impairing mitochondrial activity and inducing oxidative stress. We hypothesise that the metabolic stress occurring during serum starvation may trigger the release by donor cells of multiple bioactive factors mediating the pro-angiogenic, trophic, and antioxidant effects of their secretome.




**P-17. MESENCHYMAL STEM CELL CONDITIONED MEDIUM PROMOTES VASCULARIZATION OF BIO-COMPATIBLE SCAFFOLDS TRANSPLANTED INTO NUDE MICE**

**Ludovica Barone, Federica Rossi, Marina Borgese, Luca Buonarrivo, Mario Raspanti, Piero Antonio Zecca, Giovanni Bernardini, Rosalba Gornati**
Department of Biotechnology and Life Sciences, University of Insubria, Varese, Italy
**Objective**
Human adult mesenchymal stem cells (MSCs) have been largely studied over the past decades, for regenerative medicine applications, due to their multilineage differentiation and their potential use in several cell-based therapies. However, in the last few years, the increasing evidence showing the potential of MSC secretome has led to the acknowledgement that the use of MSC conditioned medium may represent a valid alternative to the use of stem cells, overcoming the main obstacles related to cell samples handling, survival, and rejection.
**Materials and Methods**
Accordingly, this study focuses on the characterization and in vivo application of MSC conditioned medium (CM). To this aim, MSCs have been isolated from two different sources, adipose tissue (ASCs) and dental pulp (DPSCs). Although ASCs have been largely studied, very little is known about DPSCs, therefore, DPSCs have been characterised by FACS, qPCR, and immunofluorescence up to their 30th passage to confirm their stemness maintenance over long culture. Afterward, to compare the pro-angiogenic potential of the ASC-CM and DPSC-CM vs. the cells, conditioned media, obtained after 48 h of starvation, have been mixed with a collagen scaffold, INTEGRA^®^ Flowable Wound Matrix (FWM), grafted in BALB-C nude athymic mice for 28 days and then removed, observed, and processed for gene expression and microscopy analysis. Furthermore, ASC and DPSC CMs, obtained after 72 h of starvation in both normoxic and hypoxic conditions, have been characterised by ELISA to evaluate the effect of oxygen concentration on the release of pro-angiogenic factors; then, their pro-angiogenic potential has been evaluated in vivo using the Ultimatrix sponge assay.
**Results and Conclusions**
Even though an exhaustive characterisation of the CM, which also includes the microvesicle fraction, is still in progress, the data obtained demonstrated that the scaffolds associated with CM showed the same efficiency of the ones associated with cells in promoting cellular invasion and capillary growth. Furthermore, the CMs produced under hypoxic conditions seem to promote more efficiently the angiogenesis in vivo.




**P-18. MELANOMA EXOSOMES INDUCE PD-1 OVEREXPRESSION AND TUMOUR PROGRESSION VIA MESENCHYMAL STEM CELL ONCOGENIC REPROGRAMMING**

**Edina Gyukity-Sebestyén ^1^, Mária Harmati^1^, Gabriella Dobra ^1,2^, István B Németh ^3^, Ágnes Zvara ^1^, Éva Hunyadi-Gulyás ^1^, Péter Horváth ^1,3^, Tibor Pankotai ^4^, Barbara Borsos ^4^, Miklós Erdélyi ^5^, Edit I Buzás ^6^, Lajos Kemény ^3^, Krisztina Buzás ^1,7^**




^1^ Biological Research Centre, Eötvös Lorand Research Network, Szeged, Hungary^2^ Doctoral School of Interdisciplinary Medicine, University of Szeged, Hungary^3^ Institute for Molecular Medicine Finland (FIMM), University of Helsinki, Finland^4^ Institute of Pathology, University of Szeged, Hungary^5^ Faculty of Science and Informatics, University of Szeged, Hungary^6^ Faculty of Medicine, Semmelweis University, Budapest, Hungary^7^ Department of Immunology, University of Szeged, Hungary




**Objective**
Recently, it has been described that programmed cell death protein 1 (PD-1) overexpressing melanoma cells are highly aggressive. However, until now it has not been defined which factors lead to the generation of PD-1 overexpressing subpopulations.
**Methods**
Murine primary mesenchymal stem cells (MSCs) from adipose tissue were pretreated with B16F1 melanoma cell-derived exosomes (mcde). Exosomes were stained by lipophilic dyes, and their uptake into recipient cells was visualised. The rate of apoptosis and expression of multipotent stromal cell markers were analysed by flow cytofluorometry. Tumour-bearing mice were injected with mcde-conditioned MSCs i.v.; the control mice received untreated MSCs. The whole miRNA spectra and the proteome of mcde were analysed by SOLiD5500xl technology and LC-MS/MS, respectively.
**Results**
Here, we present that melanoma-derived exosomes, conveying oncogenic molecular reprogramming, induce the formation of a melanoma-like, PD-1 overexpressing cell population (mMSC^PD-1+^) from naïve MSCs. Exosomes and mMSC^PD-1+^ cells induce tumour progression and expression of oncogenic factors in vivo. Finally, we revealed a characteristic, tumorigenic signalling network combining the upregulated molecules (e.g., PD-1, MET, RAF1, BCL2, and MTOR) and their upstream exosomal regulating proteins and miRNAs.
**Conclusions**
Our study highlights the complexity of exosomal communication during tumour progression and contributes to the detailed understanding of metastatic processes.




**P-19. CELLULAR AND STRUCTURAL CHANGES OF THE TENDON IN A RAT MODEL OF ACHILLES TENDINOPATHY: PRESENCE OF TENDON STEM/PROGENITOR CELLS AND MACROPHAGES**

**Francesca Libonati ^1^, Carlotta Perucca Orfei ^1^, Dimitrios Kouroupis ^2,3^, Enrico Ragni ^1^, Paola De Luca ^1^, Laura de Girolamo ^1^**




^1^ Laboratorio di Biotecnologie Applicate all’Ortopedia, IRCCS Ospedale Galeazzi Sant’Ambrogio, Milan, Italy^2^ Department of Orthopedics, UHealth Sports Medicine Institute, University of Miami, Miller School of Medicine, Miami, FL, USA^3^ Diabetes Research Institute & Cell Transplantation Center, University of Miami, Miller School of Medicine, Miami, FL, USA




**Objective**
Achilles tendinopathy is one of the most common tendon disorders of the lower limb in both the athletic and general population, requiring new and effective therapies. In this context, the recent evidence on the presence of tendon stem/progenitor cells (TSPCs) in tendons and the involvement of immune cells in the early stages of tendinopathy paves the way for future therapies. For this purpose, this study is intended to highlight the structural and cellular alteration occurring in a model of Achilles tendinopathy in rats in order to implement the knowledge about tendinopathy onset.
**Materials and Methods**
A total of 12 Sprague Dawley rats were treated to induce the Achilles tendinopathy by injecting type I Collagenase (3 mg/mL) in the right tendon, while the contralateral limb was either untreated (healthy control group) or saline injected (sham group). On day 7, tendon explants were histologically evaluated by 4 blinded observers using a semi-quantitative score, to highlight the structural and cellular alterations of the tissue. Immunohistochemical analysis was performed to localise TSPCs (CD90, CD146) and M1 (CD86) and M2 (CD206) macrophages within the tissue.
**Results**
The injection of type I Collagenase induced substantial structural damage with cellular alterations, disorganisation of collagen fibers, and a thickening of the paratenon region. An increase in the number of rounded cells was observed in both tendon and paratenon regions together with a higher cell density. TSPCs were more visible in paratenon than in the tendon proper, suggesting a possible aggregation of these cells in the vascularised area typically seen in a paratenon. A slight difference was observed in the presence of TSPCs between pathological and healthy tendons even if not significant. The presence of immune cells was increased in pathological tissues with visible infiltrative (M1) and resident macrophages (M2); however, there is no significant difference in the M1/M2 ratio in this model at 7 days.
**Conclusions**
Type I collagenase injection induced Achilles tendinopathy since structural changes and cellular alterations were clearly visible. As regards the inflammation and the recruitment of immune cells, the data obtained suggest their partial involvement, which should be further investigated at other timing of the tendinopathy to understand possible interaction between TSPCs and macrophages.




**P-20. COMPARATIVE ANALYSIS OF HUMAN MESENCHYMAL STROMAL CELLS DERIVED FROM ADIPOSE TISSUE AND DENTAL PULP: PHENOTYPIC CHARACTERIZATION AND SECRETOME EVALUATION FOR CELL-FREE THERAPY**

**Serena Marcozzi ^1^, Maria Assunta Ucci ^1^, Francesca Gioia Klinger ^2^, Vincenzo Campanella ^3^, Antonio Libonati ^3^, Cosimo Tudisco ^2^, Manuel Scimeca ^4^, Giulia Salvatore ^1^, Simone Vumbaca ^5^, Rosita Russo ^6^, Mario Picozza ^7^, Angela Chambery ^6^, Giovanna Borsellino ^7^, Massimo De Felici ^1^, Antonella Camaioni ^1^**




^1^ Department of Biomedicine and Prevention, University of Rome Tor Vergata, Rome, Italy^2^ Saint Camillus International, University of Health Sciences, Rome, Italy^3^ Department of Clinical Sciences and Translational Medicine, University of Rome Tor Vergata, Rome, Italy^4^ Department of Experimental Medicine, University of Rome Tor Vergata, Rome, Italy^5^ Department of Biology, University of Rome Tor Vergata, Rome, Italy^6^ Department of Environmental, Biological and Pharmaceutical Sciences and Technologies, University of Campania Luigi Vanvitelli, Caserta, Italy^7^ Neuroimmunology Unit, Santa Lucia Foundation I.R.C.C.S., Rome, Italy




**Objective**
The purpose of the present study was a comparative analysis of the biological characteristics of human Mesenchymal Stromal Cells (MSCs) derived from Adipose Tissue (ADSCs) and Dental Pulp (DPSCs) in order to evaluate their use in regenerative medicine.
**Materials and Methods**
ADSCs and DPSCs were obtained from healthy patients. ADSCs were isolated by two methods known as Stromal Vascular Fraction (SVF) and Mechanical Fragmentation (MF), while DPSCs were obtained from open apex third molars by MF after separation of the radicular (RPSCs) and the coronal region (CPSCs). These populations were compared based on their morphological features by TEM and IF, and their capacity to proliferate by WST-1 assay. Expression of surface markers was measured by flow cytometry, and the ability to undergo trilineage differentiation was also studied. Conditioned medium (CM) was collected after 3 days of culture and the Bio-Plex Pro Human Cytokine 27-plex Assay (Bio-Rad) was performed. Extracellular vesicles were separated by ExoQuick-TC and suspensions were examined by flow cytometry.
**Results**
Both ADSC cell types were able to differentiate towards adipogenic, chondrogenic, and osteogenic lineages as expected, whereas DPSC populations were unable to differentiate into adipocytes. Interestingly, the doubling time of DPSCs calculated during 96 hrs of culture was significantly lower (RPSCs = 27.72 ± 1.34 h vs. CPSCs = 35.26 ± 2.00 h) in comparison to both ADSCs (ADSCs-SVF = 69.71 ± 4.16 h; ADSCs-MF = 65.97 ± 3.75 h). In DPSCs, more than 90% were Nestin^+^ and only 10% αSMA^+^ compared with <1% and ~30% in ADSCs, respectively. Multivariate analysis by PCA for the expression of 16 surface markers measured by flow cytometry showed that organ source (dental pulp or adipose tissue) was the most critical factor that discriminates cell phenotypes. The Bio-Plex assay showed that, whereas the analyte composition of CMs from the two DPSCs appeared very similar, the isolation method was responsible for large differences among CMs from ADSCs in terms of cytokines, chemokines, and growth factors secretion. Finally, ADSCs released a significantly higher number of the smaller EVs named exosomes, ≤100 nm in diameter, than DPSCs.
**Conclusions**
These results indicate that the four MSC populations show different phenotypic and functional signatures depending on both the tissue source and the extraction method. These differences may be crucial for cell-free therapies using CMs from such cells.




**P-21. (YIA) HUMAN PLATELET LYSATE DEPLETED FROM FIBRINOGEN IS A GOOD ADDITIVE FOR LARGE-SCALE PRODUCTION OF MESENCHYMAL STEM CELLS UNDER GOOD MANUFACTURING PRACTICE CONDITIONS AND FOR LYO-SECRETOME ISOLATION, PRESERVING ITS IMMUNOMODULANT PROPERTIES**

**Elena Marini ^1^, Alessia Giovanna Santa Banche Niclot ^1^, Ivana Ferrero ^2^, Camilla Francesca Proto ^1^, Marta Barone ^1^, Giuseppe Pinnetta ^1^, Luciana Labanca ^3^, Elia Bari ^4^, Maria Luisa Torre ^4^, Katia Mareschi ^1,2^ and Franca Fagioli ^1,2^**




^1^ Department of Public Health and Paediatrics, University of Turin, Italy^2^ Stem Cell Transplantation and Cellular Therapy Laboratory, Paediatric Onco-Haematology Division, Regina Margherita Children’s Hospital, City of Health and Science of Turin, Italy^3^ Blood Component Production and Validation Centre, City of Health and Science of Turin, S. Anna Hospital, Turin, Italy^4^ University of Piemonte Orientale, Department of Pharmaceutical Sciences, Novara, Italy




**Objective**
Human Platelet Lysate (HPL) is an additive, rich in growth factors, cytokines, and proteins used to expand Mesenchymal Stem Cells (MSCs) under Good Manufacturing Practice (GMP) conditions. Preparation techniques may influence HPL composition and, therefore, the biological properties of cultured cells. Standard HPL (HPL-E) production consists of repeated freezing/thawing cycles of platelets and heparin in addition to avoiding culture medium gelling. The new method (HPL-S) consists of making platelets coagulate through the addition of Ca-Gluconate and mechanical squeezing. MSC properties are associated with their secretome and we defined a GMPprocess for freeze-dried MSC-secretome (lyo-secretome). In this work, we verified if the new HPL production method is effective and preserves the chemical and biological properties of MSCs and of their secretome.
**Materials and Methods**
From the same platelet pool, we obtained the standard HPL-E and the new HPL-S. We investigated if the treatment with Ca-Gluconate could interfere with the chemical characteristics and the growth factor amounts. Moreover, we compared the cellular growth, immunophenotype, and multipotent capacity of MSCs isolated and expanded in HPL-E and -S. We also isolated lyo-secretome from cell supernatant after serum starvation, ultrafiltration, and freeze-drying from MSCs. Lyo-secretome was analysed for lipid, protein, and growth factor contents, extra-vesicle size and concentrations, immunophenotype, anti-elastase activity, and immunomodulant properties.
**Results**
HPL-S did not contain PLTs and fibrinogen; total protein and growth factor amounts were comparable with HPL-E. The number of colony-forming unit fibroblasts showed no significant differences between the two groups. MSCs with HPL-E showed a cumulative Population Doubling higher in the earlier passages with an inversion of the growth trend in passage 4. Stem cell markers were maintained during expansion. Immunophenotypic analysis showed a significantly different expression of HLA-DR (1.30% with HPL-S, 14.10% with HPL-E) and of CD146 (>50% with HPL-S, <30% with HPL-E). Lyo-secretome obtained from MSCs with HPL-E or HPL-S did not show significant chemical/biological differences.
**Conclusions**
The use of HPL-S is an effective alternative for MSC production in GMP conditions and the obtained lyo-secretome could replace MSC-cellular therapy as a cell-free surrogate.




**P-22. BIOCOMPATIBILITY ASSESSMENT OF BIOMATERIALS OBTAINED FROM DISCARDED NATURAL SOURCES FOR DENTAL PULP STEM CELL CULTURE**

**Pasquale Marrazzo ^1^, Francesca Paris ^1^, Valeria Pizzuti ^1^, Silvia Zia ^2^, Barbara Roda ^1,2^, Francesco Alviano ^1^ and Laura Bonsi ^1^**




^1^ Department of Experimental, Diagnostic and Specialty Medicine, University of Bologna, Bologna, Italy^2^ Department of Chemistry “G. Ciamician”, University of Bologna, Bologna, Italy^3^ Stem Sel s.r.l., Bologna, Italy




**Objective**
The growth in vitro of medicinal stem cells depends on the culture method, including the attachment substrate required for adhering to cell culture. Strategies that involve the testing of new biocompatible substrates for stem cell modulation or expansion, for example, nature-inspired biomaterials, may improve the phenotype and the functionality of these cells. The chicken eggshells and honey bee pollen were usually considered waste-like materials. We aim to evaluate these two organic sources obtainable by the food industry for supporting mesenchymal stem culture.
**Materials and Methods**
Human Dental Pulp Stem Cells (DP) were selected as a mesenchymal stem cell culture model. Resazurin reduction assay was used to assess the adhesion to the eggshell membrane by living cells and, in parallel, to evaluate cytotoxicity in presence of bee pollen solution. Immunofluorescence staining was performed to display morphology of cells attached to the eggshell membrane. Monochlorobimane probe was used to evaluate glutathione levels after exposure to bee pollen.
**Results**
The viability signal of DP stem cells plated on the eggshell membrane increased according to seeded cell density. The cell attachment to the membrane was also confirmed by immunofluorescence performed on fixed cultured membranes. Bee pollen solution was able to increase metabolism-based signal of viability, as well as to increase antioxidant cellular glutathione content.
**Conclusions**
The biocompatibility of both eggshell membranes and bee pollen treatment in DP stem cells was confirmed. The experiments with both the organic materials showed positive results in terms of cell metabolism stimulation and no significant alteration of classical morphology of mesenchymal stem cells in vitro. We conclude that these data can be promising for continuing this study and starting new analyses concerning specific stem cell properties changes, in comparison to standard tissue culture plates.




**P-23. (YIA) INVESTIGATION OF THE EFFECTS OF CONTROLLED UNIAXIAL STRETCH STIMULI ON HUMAN PERIODONTAL LIGAMENT AND ADIPOSE-DERIVED STAMINAL CELLS**

**Beatrice Masante ^1,2^, Ilaria Roato ^2^, Giovanni Putame ^1^, Andrea Tancredi Lugas ^1^, Marta Tosini ^1^, Mara Terzini ^1^, Alberto Audenino ^1^, Diana Massai ^1^, Federico Mussano ^2^**




^1^ PolitoBIOMed Lab, Department of Mechanical and Aerospace Engineering, Politecnico of Turin, Italy^2^ Bone and Dental Bioengineering Lab, CIR-Dental School, Department of Surgical Sciences, University of Turin, Italy




**Objective**
The periodontal ligament (PDL) plays a key role in providing mechanical stability and absorbing the high forces associated with mastication. These capacities deteriorate if PDL is affected by periodontitis, a degenerating disease that leads to loss of the PDL and the supporting alveolar bone. In view of PDL engineering and PDL regeneration, we investigated the hPDLSCs and adipose-derived stem cells (ASCs) behaviour under controlled uniaxial stretch stimuli, exploiting a previously developed bioreactor and customized flexible substrates.
**Materials and Methods**
ASCs (ASC52 -telo hTERT, ATCC) and primary hPDLSCs were characterised for the expression of mesenchymal markers by means of a FACS analysis. The same cell types were seeded on customised flexible substrates and exposed to a controlled uniaxial stretch stimulus (15% of strain, 1 Hz, for 90 s every 6 h for 3 days, n = 3) using a previously in-house developed bioreactor. Cells cultured in static conditions were used as control (n = 3). For both cell types and both conditions, DMEM was used as a culture medium. At the end of the culture period, cells were collected and the expression of stemness markers (NANOG, SOX2, and OCT3/4) and osteogenic markers (ALP, OCN, and RUNX2) were assessed through Real-Time PCR.
**Results**
FACS analysis highlighted that the phenotypes of ASCs and hPDLSCs are comparable, both expressed CD73, CD90, CD105, and CD44, while they were negative for CD45 and HLADR. The basal stemness marker expression seems to be higher for hPDLSCs than ASCs, nonetheless, both cell types showed an increasing trend in their expression after stretch stimulation. After the stretch stimulus, the expression of the osteogenic genes was increased, especially the OCN (*p* < 0.05), in the hPDLSCs, but not in the ASCs.
**Conclusions**
The results obtained by FACS analysis demonstrated that both cell types expressed the typical mesenchymal markers, and the expression of the stemness genes showed an increasing trend after stretch stimulation. Osteogenic genes were upregulated in hPDLSCs when cultured under controlled uniaxial stretch conditions. According to these results, hPDLSCs showed more osteo differentiating ability than ASCs. Further studies are ongoing to define the response of hPDLSCs and ASCs to combined stimuli, such as mechanical (stretch) and chemical (tenogenic medium) ones, to obtain an effective system for PDL regeneration.




**P-24. 3D BRAIN ORGANOID MODELS AS A TOOL TO SCREEN FOR NUTRACEUTICALS**

**Arianna Minoia ^1^, Luca Dalle Carbonare ^1^, Jens C. Schwamborn ^2^, Silvia Bolognin ^2^ and Maria Teresa Valenti ^3^**




^1^ Department of Medicine, University of Verona, University of Verona, Italy^2^ Luxembourg Centre for Systems Biomedicine (LCSB), Developmental and Cellular Biology, University of Luxembourg, 4365 Belvaux, Luxembourg^3^ Department of Neurosciences, Biomedicine and Movement Sciences, University of Verona, Italy




**Objective**
Degenerative conditions of the skeleton and the brain are significant issues with significant socioeconomic effects. This study would underline the significance of the interaction between nerve and bone cells in 3D brain organoid models as a tool to screen for nutraceuticals which have an impact on skeletal metabolism. Knowing that one of the main and important pathways that link bone metabolism and the brain is the Wnt/β-catenin pathway and plays a crucial role in the development of many aspects of midbrain DA development, we would try to understand the effect of some molecules chronically treating 3D organoids derived from two different cell lines for up to 40 days, focusing on the study of genes involved in the Wnt/b-catenin pathway and in neuronal degeneration.
**Material and Methods**
Generation organoids: NESCs derived from iPSCs. We cultured organoids for 40 days. Sample preparation: we extract total RNA using the mRNA Qiagen-Kit. Reverse transcription was done using High-Capacity cDNA Transcription Kit. Real-time data: TaqMan probes and TaqMan Universal Master Mix to analyse the gene expression. Flow cytometry: Flow cytometry was performed using BD LSR-Fortessa. Sectioning organoids. Immunofluorescence staining. Confocal imaging
**Results**
We investigated the gene expression of PARK2, NR2F1, CTNNB1, and LRP5 for untreated 3D organoids and treated 3D organoids with DMSO, lipoic acid, and JH-II in the two cell lines.Comparing the expression of PARK2 between the organoids deriving from the Bil-WT line towards the treated organoids with mutation LRRK2-G2019S, we note a higher expression of PARK2 in the samples treated with the molecules of interest. We also found this trend of a significant increase in expression for the NR2F1 gene. As regards the LRP5 gene, we note an identification in the samples treated with lipoic acid and an increase in expression trend for the organoids treated with the JH-II inhibitor. As regards the last gene analysed, CTNNB1, there is a higher expression for the samples treated with the inhibitor of the LRRK2-G2019S mutation.
**Conclusions**
The molecules impact on the expression of the genes associated with the Wnt/β-catenin pathway upstream and downstream appear to have a beneficial impact on the rise in the expression of the target genes. Considering an improvement in the conditions of PD patients, the treatment with the molecules we examined would result in an increase in the expression of the gene of interest.




**P-25. BIOLOGICAL EVALUATION OF 3D-BIOPRINTED SCAFFOLD FOR PERIODONTAL REGENERATION**

**Alessandro Mosca Balma ^1^, Ilaria Roato ^1^, Beatrice Masante ^1,2,3^, Federico Mussano ^1^**




^1^ Department of Surgical Science, C.I.R. Dental School, University of Turin, Italy^2^ PolitoBIOMed Lab, Department of Mechanical and Aerospace Engineering, Polytechnic of Turin, Italy^3^ Interuniversity Centre for the Promotion of the 3Rs Principles in Teaching and Research, Pisa, Italy




**Objective**
Periodontitis is one of the most common diseases worldwide, causing a progressive destruction of the tooth supporting tissues. Thus, different regenerative approaches are under investigation, and this study aims to evaluate the biological properties of polycaprolactone (PCL) 3D-Bioprinted scaffolds w/or w/o alumina toughened zirconia (ATZ) filler, as suitable materials for alveolar bone and periodontal ligament (PDL) regeneration.
**Materials and Methods**
Three different types of blends were compared to each other, respectively: pure PCL, 80/20 *w*/*w* PCL/ATZ, and 60/40 *w*/*w* PCL/ATZ. The ATZ was incorporated in the PCL matrix through dissolution in chloroform by solvent casting method, then the bioink was 3D bioprinted to generate scaffolds with a standardised porous and circular geometry. ASC52 hTert cells (ASCs) were seeded on these scaffolds for 24 h to test their adhesion on the material surface. To evaluate the biocompatibility of the scaffolds, ASCs were cultured for 3, 7, and 14 days to allow scaffold colonisation, and registered their growth through the quantification of the ATP release in culture by viable cells (CellTiter-Glo kit). To investigate whether the scaffold had osteoinductive properties, ASCs were cultured in an osteogenic medium for 2 months, then RNA was extracted and the expression of osteogenic genes (ALP, COLL1, OCN, and RUNX2) were quantified with Real-Time PCR. SEM and EDX analysis were performed to evaluate the newly formed bone and to quantify the presence of calcium deposition, respectively.
**Results**
All the scaffolds allowed ASC adhesion and growth, particularly the ones with 80/20 *w*/*w* PCL/ATZ showed better biocompatibility in the long term. The addition of ATZ to the PCL matrix reduced the osteoinductive property of the materials. After two months in the osteogenic medium, the expression of ALP and RUNX2 was not different among the three polymeric scaffolds, while both COLL1 and OCN expressions were reduced in the presence of ATZ filler. These results were confirmed by SEM images and EDX analysis.
**Conclusions**
The presence of ATZ seems to reduce the osteoinductivity of the scaffolds, pointing out the importance of the material chosen as a support of new PDL regeneration, because this could limit the possibility of tooth ankylosis. Future tests will be focused on the quantification of scaffolds’ mechanical properties improvement in the presence of a toughening filler as ATZ and on tenogenic differentiation capability with PDL mesenchymal stem cells.




**P-26. HA AND PRP COMBINATIONS AS “OFF THE SHELF” DEVICE FOR CLINICAL APPLICATIONS**

**Marta Nardini ^1^, Anita Muraglia ^1,2^, Antonella D’Agostino ^4^, D’Agostino Maria ^4^, Gilberto Filaci ^1,2^, Ranieri Cancedda ^3^, Chiara Schiraldi ^4^ and Maddalena Mastrogiacomo ^1^**




^1^ Department of Internal Medicine (DIMI), University of Genoa, Italy^2^ Biotherapy Unit, IRCCS Ospedale Policlinico San Martino, Genoa, Italy^3^ Emeritus professor at the University of Genoa, Italy^4^ Department of Experimental Medicine, University of Campania “L. Vanvitelli”, Naples, Italy




**Objective**
Platelet Rich Plasma (PRP) is a well-known natural product, optimised, standardised, and used for regenerative medicine. The current goal is to identify substrates as vehicles for a gradual platelet content release. Hyaluronic acid (HA) is proposed thanks to its viscoelastic and biological properties and biocompatibility. This study aimed to set up and characterise an “off the shelf” freeze-dried and injectable device based on HA and PRP for tissue regeneration.
**Materials and Methods**
Different Molecular Weights of HA (Low and High– HA-HMW and HA-LMW) were used in combination with a standardised PRP in the ratio 1:1. Rheological analysis of compounds was performed, and the products were lyophilised. After reconstitution by water, the HA/PRP mixtures were tested in terms of cell proliferation capability and wound scratch recovery on human primary fibroblasts. With the aim of validating the methods of storage of the products, the biological activity of freeze-dried HA/PRP formulations was tested at different temperatures of conservation (25 °C, 4 °C and −20 °C) for up to 6 months.
**Results**
HA-HMW/PRP compound prompted human dermal fibroblast proliferation such as PRP alone, but the formulations needed almost half an hour for full reconstitution and strong pressure to be extruded by a 21-gauge needle. To overcome these limitations, different HA-LMW/PRP formulations were characterised and tested in terms of biological activity. These formulations are capable of sustaining cell proliferation and are stable at different temperatures and lengths of storage compared to the loss of PRP activity.Interestingly, only the lowest HA-LMW tested (56 KDa) in combination with PRP showed from one up to six months of significant preservation of the proliferation activity compared to PRP alone. This was also confirmed by in vitro scratch assay using a time-lapse video microscopy station.
**Conclusions**
In conclusion, we developed a lyophilised HA-based/PRP device that improves and preserves the PRP activity over time. HA/PRP allows the development of promising products for topical, intradermic, and intra-articular applications.




**P-27. THERAPEUTIC EFFECTS OF HUMAN PLACENTAL-DERIVED MESENCHYMAL STROMAL CELLS (PDMSCS) ON A LIPOPOLYSACCHARIDE INDUCED MOUSE MODEL OF PREECLAMPSIA (PE)**

**Anna Maria Nuzzo, Laura Moretti, Ilaria Faletti, Alberto Revelli, Alessandro Rolfo**
Dept. Surgical Sciences, University of Turin, Italy
**Objective**
Preeclampsia (PE), the most severe human pregnancy-related syndrome, is a leading cause of foetal-maternal mortality and morbidity and lack of an effective therapy. The main hallmarks of PE are severe maternal hypertension and proteinuria, expression of generalised endothelial damage, and inflammation that could lead to Foetal Growth Restriction (FGR). Human Placenta-Derived Mesenchymal Stromal Cells (hPDMSCs) are well renowned for their pro-angiogenic and anti-inflammatory effects exerted via paracrine interactions. Herein, we tested the effects of hPDMSCs-CM (Conditioned-Media) on a mouse model of preeclampsia.
**Materials and Methods**
PDMSCs were isolated from control placentae and plated (1 × 10^5^ cells/mL) in DMEM without FBS at passage 5. After 48 h, CM was collected. Preeclampsia was induced in pregnant C57BL/6NCrl mice by intravenous bacterial Lipopolysaccharide (LPS) injection. Starting from d9, maternal blood pressure and proteinuria were monitored until d19. At d11 of pregnancy, dams were injected with E.Coli LPS (1 µg/Kg). At d12, mice were randomly divided into two groups (n = 7 each) and treated intravenously as follows: plain vehicle (300 µL, placebo) and hPDMSCs-CM (300 µL, treated). At d19, mice were sacrificed. A number of foetuses, FGR, foetal reabsorption, and placental weight were evaluated. Next, placentae were processed for mRNA and protein isolation. sFlt-1, IL-6, and TNF-α gene and protein expression were evaluated by Real-Time PCR and Enzyme-Linked Immunosorbent Assay (ELISA).
**Results**
Injection of hPDMSCs-CM on d12 significantly decreased maternal systolic blood pressure and proteinuria by day 13 until term relative to placebo group. No FGR and/or reabsorbed foetuses were delivered by hPDMSCs-CM treated PE mice, while 5 FGR foetuses were found in the placebo group. No differences were found in placental weight between groups. hPDMSCs-CM injection significantly decreased sFlt-1, IL-6, and TNF-α levels in PE mice.
**Conclusions**
Our data indicate that hPDMSCs-derived trophic mediators can reverse PE-like features during pregnancy, suggesting a therapeutic role for hPDMSCs for the treatment of preeclampsia.Supported by Corion Biotech s.r.l.




**P-28. OXIDATIVE STRESS AND PLACENTAL-DERIVED MESENCHYMAL STROMAL CELLS (PDMSCS): NEW PERSPECTIVE FOR PREECLAMPSIA (PE) ETIOPATHOGENESIS**

**Anna Maria Nuzzo, Laura Moretti, Ilaria Faletti, Alberto Revelli, Alessandro Rolfo**
Dept. Surgical Sciences, University of Turin, Italy
**Objective**
Mesenchymal Stromal Cells have been highlighted as an effective antioxidant therapy. Indeed, anomalies in PDMSCs antioxidant defences might cause/contribute to the increased oxidative stress (OxS) typical of PE placentae. Herein, we compared the release of antioxidant proteins and investigated the expression of the antioxidant enzymes Catalase (CAT) and Superoxide Dismutase1 (SOD1) and of OxS-triggered cell death modulators PARP1, Caspase3, and LDOC1 in normal and PE-PDMSCs. Finally, we tested the hypothesis that PDMSCs-CM (Conditioned-Media) could restore CAT and SOD1 in H2O2-treated villous explants.
**Materials and Methods**
PDMSCs were isolated from control (n = 10) and PE (n = 10) placentae. At passage 5, cells were plated (1 × 10^5^ cells/mL) in DMEM without FBS. After 48 h, CM was collected, and total antioxidant capacity was tested by Cayman’s Antioxidant Assay. Control (n = 24) villous explants were treated for 24 h by H2O2 and next by control PDMSCs-CM for 48 h. CAT, SOD1, PARP1, Caspase3, and LDOC1 gene expression were evaluated by Real Time PCR.
**Results**
We reported a lower total antioxidant capacity (1.42 Fold Decrease) in PE-PDMSCs relative to control. We reported decreased CAT and SOD1 and increased PARP1 (*p* = 0.03), Caspase3 (*p* = 0.04), and LDOC1 (*p* = 0.05) gene levels in PE relative to control PDMSCs. After 24 h with H2O2, normal PDMSCs-CM treatment increased CAT and SOD1 and significantly decreased PARP1 (*p* = 0.02), Caspase3, and LDOC1 mRNA levels relative to controls.
**Conclusions**
Herein, we demonstrated that pathological PE-PDMSCs are characterised by aberrant antioxidant properties followed by increased production of OxS-related cell death effectors. Moreover, PDMSCs-CM promotes the expression of antioxidant enzymes, thus inhibiting OxS-mediated cell death. Indeed, our data suggest that PDMSCs-CM could be used to neutralise the exacerbated OxS typical of PE placentae, thus opening to novel PDMSCs-based therapeutic options.




**P-29. MESENCHYMAL STEM CELLS IN WOUND HEALING: FOCUS ON MSC-MEDIATED SKIN REGENERATION AFTER DERMATO-ONCOLOGICAL SURGICAL PROCEDURES**

**Alessia Paganelli, Cristina Magnoni**
UO di Chirurgia Dermatologico a Indirizzo Oncologico e Rigenerativo, AOU Policlinico di Modena, Università degli Studi di Modena e Reggio Emilia, Modena, Italy
**Objective**
To assess the role of mesenchymal stem cells (MSCs) on skin wound healing both in vitro and in vivo after dermatological surgical procedures performed for oncological purposes.
**Materials and Methods**
MSCs were obtained from discarded adipose tissue during dermato-surgical procedures. After isolation, MSCs were cultured in ascorbic-acid enriched medium in order to obtain MSC-based dermal scaffolds. Organotypic cultures were also performed to assess the pro-epithelizing properties of MSCs. MSCs were also seeded on commercially available acellular dermal matrices (ADMs) to assess whether they could exert a synergistic action. Finally, we also looked at whether MSCs were spontaneously recruited in vivo at wound sites in the presence of ADMs. Classical histology, immunofluorescence, and ELISA tests have been employed in the aforementioned experimental settings.
**Results**
MSCs can efficiently be used to produce dermal equivalents. MSCs secrete all the main components (collagen and fibronectin) of the extracellular matrix upon stimulation. MSCs guide wound re-epithelialization in vitro in the presence of keratinocytes, mainly through a paracrine action on epidermal basal stem cells. When seeded on acellular dermal collagenic substitutes, MSCs significantly increase extracellular-matrix production, therefore, confirming the potential effectiveness of combination treatments. CD90+ STRO-1+ cells were detected in the neodermis after ADM positioning, therefore suggesting efficient ADM-mediated MSC recruitment.
**Conclusions**
MSCs represent a promising tool in the regenerative setting, especially for the treatment of large surgical wounds after dermatological surgical procedures. However, further studies are needed to confirm their safety in oncological patients since a potential tumour-promoting role has recently been postulated for MSCs.




**P-30. PRODUCTION OF CLINICAL GRADE EXTRACELLULAR VESICLES (EVS) SECRETED BY MESENCHYMAL STROMAL CELLS AND INDUCED PLURIPOTENT STEM CELL-DERIVED MESENCHYMAL STROMAL CELLS FOR THE TREATMENT OF OSTEOARTHRITIS**

**Maria Elisabetta Federica Palamà ^1^, Cansu Gorgun ^1^, Georgina Shaw ^2^, Mary Murphy ^2^, Chiara Gentili ^1^**




^1^ Department of Experimental Medicine (DIMES), University of Genova, Italy^2^ Regenerative Medicine Institute (REMEDI), University of Galway, Ireland




**Objective**
Mesenchymal stromal cells (MSCs) derived extracellular vesicles (EVs) have been studied for the treatment of Osteoarthritis (OA), the most common chronic disease of the joint cartilage. A large-scale expansion of MSCs is required to meet clinical demand and this could affect the effectiveness of cells and cell products. Therefore, MSC generated from induced pluripotent stem cells (iMSC) represent a promising cellular source for the manufacture of EV therapeutics. In this study, we isolated and tested the efficacy of EV secreted by MSCs and iMSC in the treatment of OA in vitro.
**Methods**
MSCs and iMSCs were cultured in vitro in serum-free clinical grade conditions. Cells were characterised during long-term in vitro expansion for surface expression pattern, proliferation ability, senescence rate, and differentiation capacity. EVs were isolated using an FPLC-anion exchange chromatography (AEX) approach and their biological effect on IL-1α treated human chondrocytes was examined.
**Results**
The use of a serum-free, chemically defined medium for isolation and culture of hMSCs allowed us to expand a population with a stable phenotype from early to late passages. It is already well known that MSC proliferation, differentiation, and function decline with passaging. After three passages, we indeed observed a drastic impact on cell growth and differentiation. The paracrine activity of hMSCs during long-term expansion was also evaluated. The number and size of vesicles released by hMSCs increased proportionally with their age. The anti-inflammatory activity of MSC-EVs was investigated in an in vitro model on osteoarthritic chondrocytes and the expression of inflammatory cytokines such as IL-6 and IL-8 were measured. Administration of hMSC-EVs showed relevant anti-inflammatory effects only for early passages-derived vesicles (until passage three). iMSCs were also expanded for the long term to define the best culture conditions and the best time window for the isolation of EVs with maximum biological activity.
**Conclusions**
Despite the promising potential of EVs for therapeutic applications, robust manufacturing processes that would increase the consistency and scalability of EV production are still lacking. The focus of our study was directed on the determination of the optimal range of time in which MSCs and iMSC are biologically functionally in a serum-free culture system. This paracrine application may represent a novel therapeutic approach for the treatment of OA.




**P-31. MESENCHYMAL STEM CELLS COMBINED WITH ENDOTHELIAL CELLS SUPPORT SPHEROID FORMATION OF OSTEOSARCOMA CELLS**

**Micaela Pannella, Chiara Bellotti, Ania Naila Guerrieri, Toni Ibrahim, Enrico Lucarelli**
Terapie Rigenerative in Oncologia, Struttura Complessa di Osteoncologia, Sarcomi dell’osso e dei tessuti molli, e Terapie Innovative, IRCCS, Istituto Ortopedico Rizzoli, Bologna, Italy
**Background and Objectives**
The tumour microenvironment (TME) is a complex milieu that contains cancer cells and non-malignant cellular and non-cellular components which together orchestrate a complex dialog. In the last decades, several papers have proposed that mesenchymal stem cells (MSCs) play a critical role in TME formation and function. In our work, we evaluated the influence of MSCs in combination with endothelial cells on Osteosarcoma (OS) cells using multicellular spheroids. Our hypothesis is that MSCs may have a pro-tumorigenic action.
**Materials and Methods**
The OS cell lines MG-63, U-2 OS, SaOS-2, and 143B were engineered to express GFP, Endothelial Cells (HUVEC) were purchased, while MSCs were isolated at the Rizzoli. Spheroids were made by self assembly in ultra-low attachment 96 well plates. Based on previous experience in generating multicellular spheroids, we combined OS/HUVEC/MSCs with a ratio of 5:3:2.
**Results**
Commercially available OS cell lines show a different ability to grow as stable spheroids; only some of them grow rapidly as spheroids but develop necrotic cores over time. Our results revealed that the metabolic activity of cells in OS cell-only spheroids decreased with time, however, increased in hybrid spheroids with MSCs/HUVEC. OS spheroids composed of U-2 OS and SaOS-2 cells formed irregular spheroids while in the hybrid configuration, with MSCs and HUVEC, they assembled forming regular spheroids. OS spheroids composed by 143B and MG-63 cells only had a rounded and compact morphology. The hybrid spheroids of 143B cells and MSCs/HUVEC had an extremely regular and smooth surface, and over a long time, they began to form buds. The hybrid spheroids composed of MG-63 and MSCs/HUVEC had a smooth and regular surface; interestingly, after 96 h MG-63 cells started to leave the spheroid’s core and after 7 and 12 days, they formed buds. Bud formation and OS cell migration suggest an increase in OS cell motility and migration when they are cultured with MSCs/HUVEC. Several studies have shown that spheroids’ roughness is indicative of cellular invasiveness. Thus, we compared the roughness between simple and hybrid spheroids demonstrating that all hybrid spheroids exhibit greater roughness than spheroids composed of OS cells only.
**Conclusions**
Data obtained shows that combined MSCs and HUVEC support OS growth and may influence the spheroid morphology and invasiveness, thus sustaining the hypothesis of the pro-tumorigenic effect of these cells.




**P-32. DEVELOPMENT OF A PRECLINICAL MODEL OF PDAC TO INVESTIGATE MSC-BASED DELIVERY SYSTEMS**

**Smeralda Rapisarda ^1^, Benedetta Ferrara ^1^, Antonio Citro ^1^, Fabio Manenti ^1^, Chiara Gnasso ^2^, Lorenzo Piemonti ^1^**




^1^ Diabetes Research Institute, IRCCS San Raffaele, Milan, Italy^2^ Experimental Imaging Centre, IRCCS San Raffaele, Milan, Italy




**Objective**
Conventional therapies for pancreatic ductal adenocarcinoma (PDAC) present limits due to drug toxicity and the chemoresistance of PDAC cells. That can be due to the PDAC stroma, which constitutes a barrier for the transport of drugs to tumour cells and a drug-delivery system might improve the treatment. The aims of this study were firstly to develop a preclinical metastatic model of PDAC and secondly to set up a model of treatment with mesenchymal stem cells (MSCs) for future applications as a drug-delivery tool.
**Materials and Methods**
To generate the liver metastatic model, three different cell lines derived from KPC mice were used: K8484 (from PdxCre/LSL-KrasG12D-Trp53R172H), DT6606 (from PdxCre/LSL-KrasG12D), and DT6606lm (from liver metastases after intraportal injection of DT6606). Cells were injected into the portal vein of immunocompetent eight weeks C57BL/6N male mice in a dose range of 1 × 10^3^–5 × 10^5^. On day 20, after tumour induction, the metastatic growth was assessed by seven-tesla magnetic resonance (MR). On day 21, the first group of tumour-bearing mice underwent intravenous (i.v.) injection of luciferase-transduced MSCs (LUC^+^MSCs) to evaluate the MSCs biodistribution using in vivo imaging system (IVIS). A second group of tumour-bearing mice underwent an intraportal injection of LUC^+^MSCs and the LUC signal intensity of the two groups was compared.
**Results**
Among the three investigated cell lines, only the K8484 cell line was chosen to generate the metastatic model. K8484 cell-derived liver metastases, indeed, exhibited a largely glandular architecture, more similar to human metastases. Moreover, the use of the intraportal model allowed a homogeneous and synchronous metastatic growth in mice 20 days after the injection. Studies on biodistribution of i.v. injected MSCs revealed a cell accumulation in the lung after injection with a low-intensity signal and several animals died after cell administration. The mice subjected to intraportal injection of MSCs reported a high cell accumulation in the liver with a prolonged signal up to six days after injection, making the intraportal injection preferable to allow MSCs to reach the tumour site.
**Conclusions**
The development of this metastatic model of PDAC in the liver allowed us to evaluate MSCs as a promising therapeutic option for PDAC, especially using intraportal injection to deliver them directly to the tumour site. Obtained results may open up new insights into the use of MSCs as tools to treat PDAC.




**P-33. HUMAN AMNIOTIC MESENCHYMAL STROMAL CELL CONDITIONED MEDIUM MODIFIES CANCER ASSOCIATED FIBROBLAST GENE EXPRESSION AND CYTOSKELETAL ORGANIZATION**

**Jacopo Romoli ^1^, Andrea Papait ^1,2^, Paola Chiodelli ^3^, Patrizia Bonassi ^3^, Silvia De Munari ^3^, Elsa Vertua ^3^, Serafina Farigu ^3^, Antonietta Silini ^3^, Ornella Parolini ^1,2^**




^1^ Department of Life Science and Public Health, Università Cattolica del Sacro Cuore, Rome, Italy^2^ Fondazione Policlinico Universitario “Agostino Gemelli” IRCCS, Rome, Italy^3^ Centro di Ricerca E. Menni, Fondazione Poliambulanza Istituto Ospedaliero, Brescia, Italy




**Objective**
Tumour-associated stromal cells, also known as carcinoma-associated fibroblasts (CAFs), play a pivotal role in favouring tumour growth by their interactions with both tumour cells and cells present in the tumour microenvironment (TME). Recent studies demonstrate that CAFs can favour metastasis and are responsible for chemo-resistance mechanisms. Human amniotic mesenchymal stromal cells (hAMSC) are known to exert immune-modulatory and anti-fibrotic effects, targeting not only immune cells but also stromal cells in damaged tissues. The aim of our study was to determine if hAMSC could also exert effects on stromal cells within the tumour microenvironment, namely CAFs. To this end, we in vitro stimulated the differentiation of normal adult fibroblasts towards CAFs with exogenous TGF-β. The acquisition of CAF features was assessed by immunofluorescence for αSMA expression and for cytoskeleton organisation. In addition, we used Real-Time PCR and flow cytometry to assess the expression of the most relevant CAF markers. Finally, we evaluated the ability of hAMSC conditioned medium (CM-hAMSC) to counteract the acquisition of CAF-like gene expression and functionality.
**Materials and Methods**
Conditioned medium from hAMSC was obtained by culturing hAMSC at passage 2 in DMEM-F12 without serum for 5 days. Human dermal fibroblasts were treated with TGF-β ± CM-hAMSC for 3, 7, and 11 days. At each timepoint, flow cytometry, immunofluorescence, and RT-PCR were performed to assess CAF-like phenotype.
**Results**
Flow cytometry showed a high percentage of αSMA^+^ cells after TGF-β treatment, which decreased after CM-hAMSC treatment. Concurrently TGF-β-treated fibroblasts displayed well-organised cytoskeletal αSMA and presented extracellular MFAP5 deposition, whereas CM-hAMSC treatment interfered with these processes. These data are supported by RT-PCR analysis which showed αSMA and MFAP5 downregulation when dermal fibroblasts were treated with CM-hAMSC. On the other hand, we observed an increase of PDPN gene expression and positive cells after CM-hAMSC treatment.
**Conclusions**
Our preliminary results suggest that hAMSC-secreted factors are able to inhibit CAF activation. Further studies will be aimed at confirming these results and understanding if hAMSC-secreted factors inhibit CAF migration and ultimately clarify the use of hAMSC as a potential therapeutic strategy able to not only target tumour cells, but also stromal cells within the tumour microenvironment.




**P-34. MESENCHYMAL STROMAL CELLS FOR THE TREATMENT OF CRANIAL CRUCIATE LIGAMENT RUPTURE OF DOGS**

**Gabriele Scattini ^1^, Piero Boni ^2^, Alessandro Fruganti ^3^, Fabrizio Dini ^3^, Luisa Pascucci ^1^**




^1^ University of Perugia, Department of Veterinary Medicine, Perugia, Italy^2^ Veterinary practitioner, Cannara (PG), Italy^3^ University of Camerino, School of Biosciences and Veterinary Medicine, Matelica, Italy




**Objective**
In recent years, mesenchymal stromal cells (MSC) have received a strong boost in veterinary medicine due to their pro-regenerative properties. MSCs are virtually present in all the organs possessing a vascular stroma. The aim of this study was to evaluate the potential therapeutic use of MSC from adipose tissue in dogs with complete cranial cruciate ligament (CCL) ruptures that had not undergone surgical treatment. CCL rupture is the most common cause of lameness in dogs and can be treated conservatively, but surgical therapy is the treatment of choice to restore stability and functionality of the joint. However, surgery does not prevent the development of osteoarthritis, which over time leads to relapse of lameness and pain. It is worth mentioning that an increasing number of public and private veterinary hospitals all over the world treat partial CCL rupture with MSC as a second-line or even first-line therapy. There is no evidence of MSC use in complete ruptures.
**Material and Methods**
Tyson, an American Staffordshire Terrier, male, 4 years old, received a diagnosis of complete CCL rupture followed by 4 intra-articular infiltrations of autologous MSC. Post-infiltration monitoring was performed by orthopaedic, ultrasound, radiography examination, and nuclear magnetic resonance (NMR). A 20-month follow up was performed.
**Results**
MSC did not trigger adverse effects in the short to medium term, nor they cause regeneration of the CCL. However, they displayed a strong anti-inflammatory activity responsible for symptom remission. The therapeutic effectiveness of MSC seems to be attributable to their chondroprotective effect which slows down the development and progression of osteoarthritis.
**Conclusions**
The results described in this study suggest that the use of MSC for the treatment of complete rupture of the CCL could represent a valid option: (i) As an alternative to surgery when the patient or owners are unable to deal with it; (ii) As an adjuvant to surgical therapy both in the peri-operative and in the post-operative in order to obtain a stabilisation and reduce the development of osteoarthritis; (iii) as a substitute of non-steroidal anti-inflammatory drugs with the advantage of reducing side effects of pharmacological therapy.




**P-35. MIR-214 MEDIATED STROMA-TUMOUR CELL CROSSTALK DURING TUMOUR PROGRESSION**

**Francesca Orso ^1,2^, Federico Virga ^1,2,10^, Daniela Dettori ^1,2^, Alessandra Dalmasso ^1,2^, Mladen Paradzik ^1,2^, Aurora Savino ^1,2^, Stefania Cucinelli ^1,2^, Maura Coco ^1,2^, Iris Chiara Salaroglio ^3^, Joanna Kopecka ^3^, Margherita Aalba Carlotta Pomatto ^4^, Giovanni Camussi ^4^, Katia Mareschi ^5,6^, Leonardo Salmena ^7^, Paolo Provero ^8,9^, Valeria Poli ^1,2^, Chiara Riganti ^3^, Massimiliano Mazzone ^1,2,10^ Pier Paolo Pandolfi ^1,2,11^, Daniela Taverna ^1,2^**




^1^ Molecular Biotechnology Center (MBC), Turin, Italy^2^ Dept. Molecular Biotechnology and Health Sciences, University of Turin, Italy^3^ Department of Oncology, University of Turin, Italy^4^ Department of Medical Sciences, University of Turin, Italy^5^ Paediatric Onco-Haematology Division, Regina Margherita Children’s Hospital, City of Health and Science of Turin, Italy^6^ Department of Public Health and Paediatrics, University of Turin, Italy^7^ Princess Margaret Cancer Centre, University Health Network, ON, Canada^8^ Centre for Omics Sciences, IRCCS San Raffaele Scientific Institute, Milan, Italy^9^ Department of Neurosciences “Rita Levi Montalcini”, University of Turin, Italy^10^ Center for Cancer Biology (CCB), VIB, Leuven, Belgium^11^ Renown Institute for Cancer, Nevada System of Higher Education, Reno, NV, USA




**Objective**
Cancer and stroma cells continuously interact during tumour progression and influence each other. Secreted microRNAs (miRNAs) have recently been implicated in the tumour–stroma crosstalk. Here, we show that miR-214 is highly expressed in stromal cells such as Mesenchymal Stem Cells (MSCs) or Cancer-Associated Fibroblasts (CAFs), often derived from CAFs, and that it correlates with stromal signatures in human breast cancers and melanomas. Upon tumour cell signals, stroma miR-214 is released via Extracellular Vesicles (EVs) and is instrumental for cancer cells to promote metastasis formation through the activation of a pro-metastatic pathway which involves the protein-coding genes TFAP2C, ITGA5, and ALCAM and the anti-metastatic small non-coding RNA, miR-148b. Metabolic rewiring and, particularly, reprogrammed glucose metabolism is a hallmark of cancer, crucial for tumour progression.
**Material and Methods**
We analysed the impact of stroma-derived miR-214 on the metabolic *status* of tumour cells and observed glycolysis enhancement and an Oxidative Phosphorylation (OXPHOS) impairment linked to metastatic traits.
**Results and Conclusions**
Our results underline the relevance of “stroma miR-214” for tumour dissemination and metastasis formation and suggest the possibility of a double-edge therapeutic approach based on the targeting of miR-214 and of major metabolic players in tumour and/or stroma cells.




**P-36. CELLULAR SENESCENCE IN SYNOVIAL FLUID MESENCHYMAL STROMAL CELLS AND POTENTIAL IMPLICATIONS IN THE PROGRESSION OF OSTEOARTHRITIS**

**Gabriella Teti ^1^, Valentina Gatta ^1^, Francesca Chiarini ^2^, Mirella Falconi ^3^**




^1^ Department of Biomedical and Neuromotor Sciences, University di Bologna, Italy^2^ Dipartimento di Scienze Biomediche, Metaboliche e Neuroscienze, University of Bologna, Italy^3^ Department of Experimental, Diagnostic and Specialty Medicine—DIMES, Bologna, Italy




**Objective**
Osteoarthritis (OA) is characterised by cartilage degradation, joint inflammation, subchondral bone remodelling, and fibrosis. Current therapies for OA are mainly focused on treating symptoms of pain rather than to counteract the progression of the disease. Recently, OA has been associated with the accumulation of senescent cells in joint tissues but how senescence can affect each resident joint cell and its link with the progression of OA are still poorly described. With the advance of tissue engineering and regenerative medicine, mesenchymal stem/stromal cells (MSCs) have represented a promising candidate for cartilage repair and regeneration. However, most of the current research has been focused on the application of MSCs derived from bone marrow, and just a few studies have investigated the potential therapeutic properties of MSCs isolated from synovial fluid (sf-MSCs). Although sf-MSCs have demonstrated higher chondrogenic capabilities their accumulation as senescent cells in synovial fluid and their correlation with OA progression has never been investigated. Thus, the aim of the study was to verify the presence of senescent sf-MSCs isolated from OA joints and compare their chondrogenic capabilities with sf- MSCs isolated from healthy joints.
**Material and Methods**
Sf-MSCs were isolated from tibia-tarsal joints of healthy and diseased horses with an established diagnosis of OA. Cells were cultured in vitro and characterised for cell proliferation assay, cell cycle analysis, ROS detection assay, ultrastructure analysis, and evaluation of senescent markers. To evaluate the influence of OA on chondrogenic differentiation, sf-MSCs isolated from diseased joints were in vitro stimulated to chondrogenic differentiation and the expression of chondrogenic markers was checked and compared to healthy sf-MSCs.
**Results**
Results clearly showed an arrest of the cell cycle in combination with upregulation of the senescent markers p21^WAF1/Cip1^ and p16^INK4^, reduced autophagy and increased ROS production in OA sf-MSCs, demonstrating a senescent state. Furthermore, OA sf-MSCs showed a reduced ability in synthesising proteoglycans while the ability to express collagen II is upregulated, suggesting the involvement of OA sf-MSCs in developing a fibrotic joint environment.
**Conclusions**
In conclusion, our results demonstrate the presence of senescent sf-MSCs in OA joints which could have a key role in the progression of the disorder.




**P-37. LASER DISSECTION-COUPLED QUANTITATIVE MICROLIPIDOMIC METHOD TO RESOLVE TUMOUR HETEROGENEITY**

**Vanda Varga-Zsíros ^1,2^, Mária Péter ^1^, Ede Migh ^1^, Annamária Marton ^1^, Aladár Pettkó-Szandtner ^3^, Imre Gombos ^1^, Zoltán Kóta ^4^, Péter Horváth ^1^, László Tiszlavicz ^5^, Zsuzsanna Darula ^3,4^, Csaba Vizler ^1^, Zsolt Török ^1^, László Vígh ^1^, Gábor Balogh ^1^**




^1^ Biological Research Centre, Institute of Biochemistry, ELKH, Szeged, Hungary^2^ University of Szeged, Ph.D. School in Biology, Szeged, Hungary^3^ Biological Research Centre, Laboratory of Proteomics Research, Szeged, Hungary^4^ Single Cell Omics Advanced Core Facility, HCEMM, Szeged, Hungary^5^ University of Szeged, Department of Pathology, Szeged, Hungary




**Objective**
Lipid metabolic reprogramming is a newly recognized hallmark of malignancy. Most normal cells build up their membranes from dietary lipids. In contrast, cancer cells reactivate the de novo lipogenesis. To understand the aggressiveness and metastatic potential of tumours, the exploration of tumour heterogeneity is of great interest but also a great challenge.
**Materials and Methods**
We developed and validated a novel laser microdissection-coupled multi omics platform which combines quantitative and comprehensive lipidome, proteome, and/or transcriptome analysis with spatial resolution. The multistep approach involves the preparation of parallel cryosections from spheroids or different tissue samples, cross-referencing of hematoxylin–eosin-stained and native images, laser microdissection of marked regions (down to ~15 cells), in situ lipid microextraction/protein digestion or RNA isolation, and high-performance mass spectrometry (direct injection-based shotgun lipidomics and UPLC-coupled proteomics on an Orbitrap Lumos instrument) or transcriptomics.
**Results**
We identified a radial gradient in the lipidomic profile of 4T1 mouse breast cancer spheroids which correlated well with nutrient availability. By using mouse allografts injected with 4T1 cells, we observed substantially different lipidomic patterns not only between tumorous and non-tumorous areas of the liver or spleen but also between tumorous regions grown in different microenvironments. Close-distant parallels of cryosections were subjected to other omics and/or staining procedures.
**Conclusions**
The integration of lipidomic results with transcriptomic, proteomic, and immunohistological data can improve our understanding in tumour heterogeneity as well as in the pathology of various lipid-related disorders.


## 6. Summary

This conference report reflects the consensus viewpoint of the authors and scientists participating in the 2022 GISM annual meeting: Tarlan Eslami Arshaghi, Paola Bagnoli, Laura Barrachina Porcar, Luca Battistelli, Anna Brini, Stefania Bruno, Elena Ceccotti, Lucia Ceresa, Valentina Coccè, Stefano Cosma, Nicholas Crippa Orlandi, Maurizio Del Bue, Laura De Girolamo, Silvia Dotti, Roisin Dwyer, Franca Fagioli, Andrea Grosso, Maria Harmati, Ana Ivanoska, Enrico Lucarelli, Maddalena Mastrogiacomo, Barbara Merlo, Matteo Moretti, Marta Nardini, Andrea Papait, Graziella Pellegrini, Augusto Pessina, Michela Pozzobon, Enrico Ragni, Ilaria Roato, Silvia Scaglione, Gabriele Scattini, Maria Luisa Torre, Roberta Visone, and Silvia Zia.

## Figures and Tables

**Figure 1 ijms-24-08902-f001:**
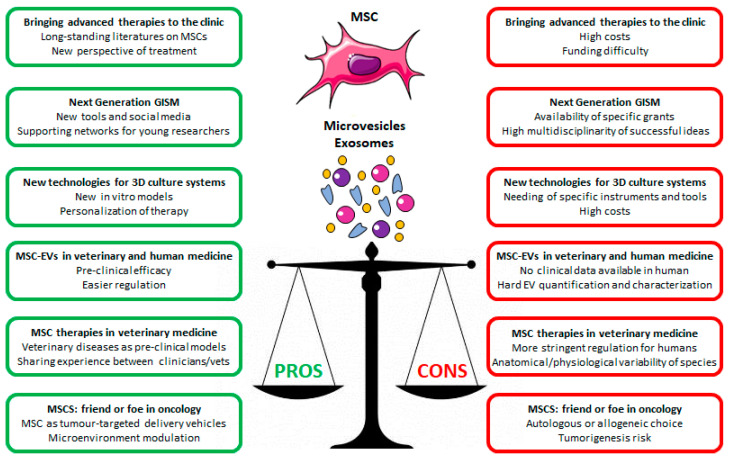
Pros and cons in mesenchymal stem cells utilisation, as raised from the six sections of the congress.

## Data Availability

Not applicable.

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
