# Peer review of "State of the Art and New Trends from the 2022 Gism Annual Meeting"

_ijms, 2023, doi:10.3390/ijms24108902_

Round 1
Reviewer 1 Report
This manuscript is a conference report of the 2022 Italian Mesenchymal Stem Cell Group (Gruppo Italiano Staminali Mesenchimali, GISM) Annual Meeting.
The authors summarized very well about topics of the conference and adequately described the importance of topics which were presented in the meeting. There are many interesting topics in this report. This conference report can give novel insight to the researchers and scientists who are working with mesenchymal stem cells.
Author Response
Thank you very much for your positive comments
Reviewer 2 Report
The Conference report STATE OF THE ART AND NEW TRENDS FROM THE 2022 GISM ANNUAL MEETING by Ivana Ferrero, Camilla Francesca Proto, Alessia Giovanna Santa Banche Niclot, Elena Marini, Luisa Pascucci, Filippo Piccinini, Katia Mareschi, summarize new soundness works in Mesenchymal stem cells field. I believe that this report is really important and well done as it offers the reader new food for thought, as well as stimulating the design of new ideas despite not having participated in the congress.
An added value is represented by the fact that the function and impact of mesenchymal cells has been divided into six sections (1) Bringing advanced therapies to the clinic: trends & strategies, 2) GISM-Next Generation, 3) New technologies for 3D culture systems, 4) Therapeutic applications of MSC-EVs in veterinary and human medicine, 5) Advancing MSC therapies in veterinary medicine: present challenges and future perspectives, 6) MSCs: a double-edged sword: friend or foe in oncology), this helps the reader to position himself in his own field but also to mature ideas based on interdisciplinarity
However, I ask the authors to implement the conclusions for each section by emphasizing more the nodal points and identifying a common perspective with respect to all interventions.
In addition, the authors could create a cartoon that summarizes the pros and cons in mesenchymal stem cells utilization in the six sections analyzed, perhaps in a perspective of One Health.
Minor editing of English language required
Author Response
Thank you very much for your comments and suggestions.
We have implemented the conclusions as you have suggested and inserted a cartoon that summarizes the pros and cons in mesenchymal stem cells utilization in the six sections analysed
Reviewer 3 Report
nothing specific
Author Response

(The authors gave the same response as above.)
